# Learning To Draft: Adaptive Speculative Decoding with Reinforcement Learning

**Jiebin Zhang**[♠♡*]   **Zhenghan Yu**[♠♡]   **Liang Wang**[♢]   **Nan Yang**[♢]   **Eugene J. Yu**[♠]   **Zheng Li**[♠]
**Yifan Song**[♠]   **Dawei Zhu**[♠]   **Xingxing Zhang**[♢]   **Furu Wei**[♢]   **Sujian Li**[♠♡]
[♠]National Key Laboratory for Multimedia Information Processing, Peking University
[♡]School of Computer Science, Peking University
[♢]Microsoft Research Asia
{zhangjiebin,lisujian}@pku.edu.cn   wangliang@microsoft.com
https://github.com/zhzihao/Learning-to-Draft

## Abstract

Speculative decoding accelerates large language model (LLM) inference by using a small draft model to generate candidate tokens for a larger target model to verify. The efficacy of this technique hinges on the trade-off between the time spent on drafting candidates and verifying them. However, current state-of-the-art methods rely on a static time allocation, while recent dynamic approaches optimize for proxy metrics like acceptance length, often neglecting the true time cost and treating the drafting and verification phases in isolation. To address these limitations, we introduce Learning to Draft (LTD), a novel method that directly optimizes for throughput of each draft-and-verify cycle. We formulate the problem as a reinforcement learning environment and train two co-adaptive policies to dynamically coordinate the draft and verification phases. This encourages the policies to adapt to each other and explicitly maximize decoding efficiency. We conducted extensive evaluations on five diverse LLMs and four distinct tasks. Our results show that LTD achieves speedup ratios ranging from $2.24\times$ to $4.32\times$, outperforming the state-of-the-art method Eagle3 up to 36.4%.

## 1 Introduction

Modern large language models (LLMs) are increasingly employed to perform deliberate, multi-step reasoning to solve complex tasks, a paradigm that extends their performance capabilities at inference time (Guo et al., 2025; Team, 2025b; Agarwal et al., 2025). However, the complexity of these reasoning processes leads to significant latency, which has prompted a growing research focus on accelerating inference while preserving high-quality output (Sui et al., 2025). Among these efforts, speculative decoding stands out as a highly effective solution, using a smaller, faster draft model to generate token sequences that are then verified in parallel by the larger target model. This approach ensures both accelerated inference and a preserved output distribution (Leviathan et al., 2023).

The quality and structure of the draft token sequences ("drafts" for short) are crucial to the overall performance of speculative decoding. Early methods primarily employ simple chain-structured drafts (Chen et al., 2023; Leviathan et al., 2023). In contrast, recent studies have demonstrated that tree-structured drafts can significantly increase the number of accepted tokens (acceptance length) per draft–and-verify cycle (Miao et al., 2023; Li et al., 2024; Cai et al., 2024). State-of-the-art methods, such as Eagle3 (Li et al., 2025a), employs a static, one-size-fits-all strategy, configured through manual tuning, to define the size of the draft trees. While some works have attempted to dynamically adjust the size of the draft tree (Brown et al., 2024; Zhang et al., 2024c; Huo et al., 2025; Huang et al., 2025a), their approaches typically rely on simple heuristics or train models to optimize for the metric of acceptance length—the average number of tokens guessed correctly per verification step. However, maximizing acceptance length does not necessarily lead to faster overall generation speed, as it completely ignores the critical time cost of the drafting and verification processes themselves. The true objective should be maximizing the model's throughput. Otherwise, the pursuit of a longer

---

* Contribution during Jiebin's internship at MSR Asia. Sujian Li is the corresponding author.

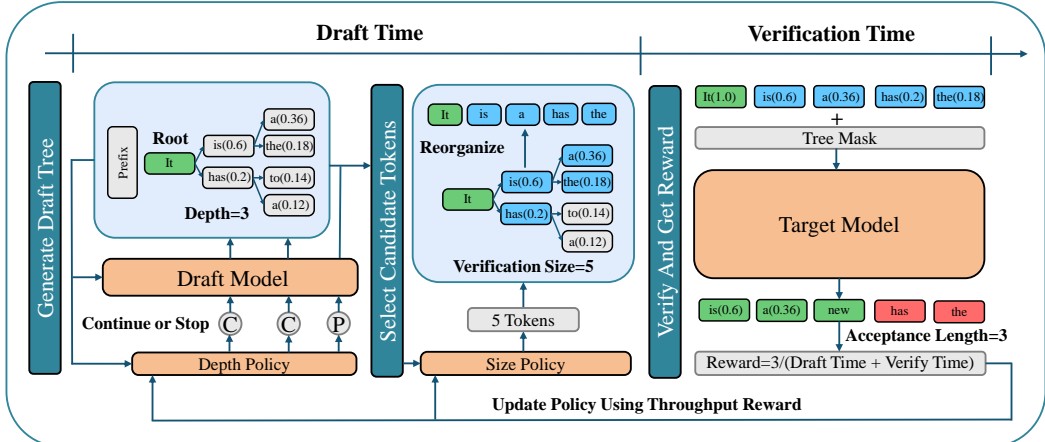

Figure 1: The overview of LTD method. We formulate the interaction between draft model and target model as an RL environment. We employ two policies to dynamically manage the allocation between draft generation and target verification by controlling the draft tree's depth and verification size. These policies are optimized together using the reward signal based on the throughput of each draft-and-verify cycle.

acceptance length can result in a large draft tree whose generation and verification consume more time than is saved, ultimately slowing down the process. At the same time, one key limitation identified in existing methods is the isolated optimization of drafting and verifying draft trees.

In this paper, we argue that drafting and verification are interdependent and should be co-optimized. A truly optimal strategy must be both time-aware and adaptive to the interplay between the two stages. For example, in a simple text generation scenario, if only the drafting stage is optimized while the verification stage remains fixed, a suboptimal outcome occurs. Even if the draft model can easily generate a long and correct sequence, it is forced to truncate its output to accommodate the limited verification cost. This leads to a waste of drafting capacity, preventing it from reaching its full potential for speedup. In an ideal joint optimization, the draft stage would produce a longer and more complete sequence, aware that the verification stage will dynamically adjust its resources. In turn, informed of the long drafts, the verification stage would allocate sufficient computational capacity to validate these tokens efficiently.

Based the idea above, we propose LTD (learning to draft), a novel method that synergistically optimizes both phases by using throughput as the direct optimization objective. Specifically, we formulate the interaction between the draft and target models as a reinforcement learning (RL) environment. As shown in Figure 1, we train two distinct, co-adaptive policies: a depth policy to control draft cost by determining the draft tree's depth, and a size policy that manages verification cost by selecting the optimal number of candidate tokens for the target model. Crucially, these policies are trained using the throughput of each draft-and-verify cycle as a direct reward signal. Furthermore, we train these two policies iteratively, which allows them to co-adapt to each other's evolving strategies. These enable them to learn how to adaptively manage the trade-off between acceptance length and time cost, and dynamically coordinate the two phases to maximize inference efficiency.

We build our method upon the state-of-the-art Eagle3 framework, training our policies to replace its static heuristics with a dynamic, context-aware decision-making process. As demonstrated through extensive evaluations under five distinct LLMs across four benchmarks, our method achieves significant performance gains. During greedy decoding, our method yields substantial throughput improvements over the highly optimized Eagle3 baseline: 36.4% on Qwen3-32B, 9.5% on Deepseek-Distilled-Llama-8B, 6.5% on Llama-3-8B, 5% on Vicuna-1.3-13B, and 4% on Qwen3-14B. Notably, our approach demonstrates robustness in high-temperature scenarios, delivering an approximate 5% throughput gain where most other dynamic speculative methods become ineffective. Our contributions can be summarized as follows:

- Direct throughput optimization: We propose a time-aware optimization approach by formulating the speculative decoding process as an RL environment. By maximizing throughput of each draft-and-verify cycle as a direct reward, our method learns to manage the trade-off between acceptance length and time cost.
- Co-adaptive policy framework: Instead of treating the drafting and verification phases as independent problems, we propose a jointly trained, co-adaptive policy framework where two policies learn to dynamically coordinate to each phase.
- SOTA performance and robustness: Our method achieves significant performance gains (up to 36.4%) over strong Eagle3 baseline in greedy decoding. Compared to prior work, it maintains its effectiveness in high-temperature sampling scenarios.

## 2 RELATED WORK

### 2.1 SPECULATIVE DECODING

Vanilla Speculative Decoding uses a chain-like structure to generate token candidate tokens (Stern et al., 2018; Chen et al., 2023; Leviathan et al., 2023). Moving beyond these initial chain-structured approaches, SpecInfer (Miao et al., 2023) introduced a tree-based drafting architecture. By incorporating a tree attention mechanism, this method facilitates the parallel verification of multiple candidate sequences, establishing a highly efficient paradigm that has since been widely adopted (Cai et al., 2024; Li et al., 2024). Subsequent research has predominantly focused on refining the draft model through several key strategies, such as leveraging knowledge distillation (Liu et al., 2023; Zhou et al., 2023), utilizing more effective features for draft token prediction (Cai et al., 2024; Li et al., 2025a; Ankner et al., 2024; Zhang et al., 2024a; Li et al., 2025b; Du et al., 2024), and employing self-speculation, wherein components of the target model are repurposed for drafting (Zhang et al., 2023; Elhoushi et al., 2024; Hooper et al., 2025).

### 2.2 ADAPTIVE ADJUSTMENTS FOR SPECULATIVE DECODING

While tree-based verification significantly enhances throughput, most methods adopt a static schedule for each draft-and-verify cycle (Cai et al., 2024; Li et al., 2025a), failing to dynamically adapt to the varying complexities of input contexts. Recognizing this limitation, recent efforts have increasingly focused on dynamic methods that adapt the speculative strategy based on the generation context. These works can be broadly categorized into two main streams: dynamically adjusting the draft time by controlling generation length or draft depth (Brown et al., 2024; Mamou et al., 2024; Zhang et al., 2024c;b; Huang et al., 2025a; Gautam et al., 2025), or dynamically adjusting the verification time by selecting candidate tokens for target models (Zhong et al., 2024; Wang et al., 2025; Huo et al., 2025). These approaches typically leverage contextual information such as token probabilities (Brown et al., 2024), entropy (Mamou et al., 2024), or hidden states (Zhang et al., 2024b; Huang et al., 2024) to adjust speculative decoding with the goal of maximizing the expected acceptance length. Further research explored other dimensions for optimization. Some methods focus on adjusting the width of the draft tree (Qin et al., 2025; Xiong et al., 2025). Other works explore dynamic methods for Self-Speculative Decoding (Zarch et al., 2025). Research has also targeted the improvement of overall Service Level Objectives (Huang et al., 2025b) or the optimization of parallel execution between the drafting and verification stages (Liu et al., 2025).

## 3 MODEL OVERVIEW

### 3.1 PRELIMINARY STUDY: TIME COST AND ACCEPTANCE LENGTH

We conducted preliminary experiments on the Llama and Vicuna models to validate our idea: that dynamic adjustment of the key parameters (draft depth and verification size) is necessary in draft-and-verify inference to achieve optimal acceleration by maximizing throughput. Here, the throughput for each draft-and-verify cycle ($\lambda_c$) is defined as the number of accepted tokens ($L_A$) divided by the total wall-clock time of the draft-and-verify cycle ($T_{\text{total}}$):

$$\lambda_c = L_A / T_{\text{total}} \tag{1}$$

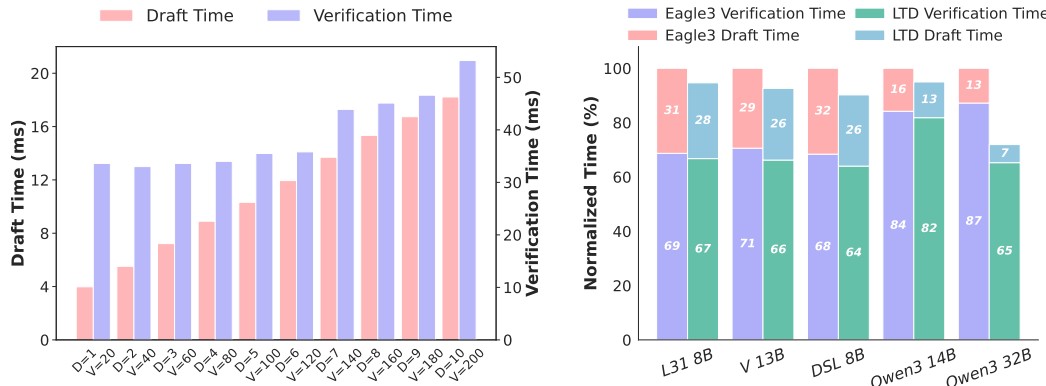

Figure 2: Analysis of Draft Time and Verification Time. Left: The average draft and verification latency of Vicuna-13B as a function of draft depth D and verification size V on benchmarks. Right: The composition of the total inference time for Eagle3 and LTD on the benchmarks, with Eagle3's total time normalized to 100% as baselines.

$T_{\text{total}}$ can be decomposed into the draft time, $T_{\text{draft}}$, and the verification time, $T_{\text{verify}}$:

$$T_{\text{total}} = T_{\text{draft}} + T_{\text{verify}} \tag{2}$$

The draft time $T_{\text{draft}}$ is primarily determined by the draft depth ($D$), which refers to the number of forward passes performed sequentially by the draft model (i.e., the steps of generating candidate tokens). A larger $D$ leads to more candidate tokens being generated, but it also results in a linear increase in the required time, as illustrated by the red bars in Figure 2 (Left). Conversely, the verification time is mainly governed by the verification size ($V$), which is the total number of candidate tokens processed in one forward pass of the large target model. Due to the model's computational cost and the long input sequence, the verification time exhibits a step-wise increase as $V$ grows, as shown by the blue bars in Figure 2 (Left). $V$ can be substantial (e.g., up to 1000 tokens when $D = 10$), making it a critical factor for verification time. With our LTD strategy, the verification time can be significantly reduced, as shown Figure 2 (Right).

Besides time latency, Figure 5 also indicates that draft depth and verification size critically affect the average acceptance length, given a fixed draft model and target model (see Appendix). Increasing both draft depth ($D$) and verification size ($V$) can generate more candidate tokens, which has the potential to raise acceptance length per cycle ($L_A$, the numerator in the throughput equation). However, increasing $D$ and $V$ inevitably leads to a rise in both draft time and verification time (i.e., an increase in the denominator, $T_{\text{total}}$). Therefore, our goal is not to blindly maximize $D$ and $V$, but rather to find an optimal balance that maximizes the overall throughput, defined as $\lambda_c = L_A/T_{\text{total}}$.

## 3.2 DRAFT-AND-VERIFY CYCLE

Our work builds upon the state-of-the-art tree-based approach, namely Eagle3 (Li et al., 2025a). In this section, we formally define the mechanics of a single draft-and-verify cycle and the role of our policies in guiding this process. As shown in Figure 1, the draft-and-verify cycle can be divided into three parts: Draft Tree Generation, Candidate Token Selection and Draft Tree Verification.

**Draft Tree Generation** The drafting process begins after the target model has output the token following the last accepted one; this token serves as the root for a new draft tree. The tree is constructed through an iterative, beam-search-like expansion process governed by two key hyper-parameters: the beam width, $W$, and the draft depth, $D$. The construction process commences with an initial expansion from the root token. In this first step, the draft model performs a forward pass to generate a probability distribution over the subsequent tokens, and the top-$W$ most probable candidate tokens are selected to form the first level of leaf nodes in the tree. Following this, the tree undergoes iterative growth for $D - 2$ subsequent rounds. In each round, the draft model takes the $W$ leaf nodes from the previous level as inputs. For each of these $W$ nodes, it generates $W$ new child

candidates, creating a combined pool of $W^2$ new nodes. A beam selection is then performed on this pool, where the top-$W$ candidates with the highest predicted cumulative path probabilities are chosen to become the new leaves of the tree, poised for the next round of expansion. The draft depth for Eagle3 is set default to 8, while in LTD our depth policy will decide whether to continue or stop after each draft model's forward pass. Once the depth policy decide to stop the expansion, the whole draft tree will pass to the size policy for candidate token selection.

**Candidate Token Selection**   With a draft depth of $D$, the final draft tree contains a total of $V_{all} = 1 + W + (D - 2) \cdot W^2$ candidate tokens, each with an associated predicted probability. Once the draft tree is generated, a subset of $V$ candidate tokens with the highest overall probabilities is selected for verification. While Eagle3 employs a fixed verification size $V$, LTD utilizes a size policy to dynamically determine this value based on the properties of the generated draft tree. These $V$ selected tokens are then flattened into a single sequence, along with a corresponding tree attention mask, is passed to the target model.

**Draft Tree Verification**   The target model performs a single forward pass to validate these $V$ tokens in parallel. It accepts the sequence of tokens up to the first point of mismatch between its own prediction and the candidate tokens. At the position of this first error, target model discards the remainder of the candidate tokens, computes the correct token, and appends it to the sequence of accepted tokens. This newly confirmed token then becomes the root of draft tree for the next draft-and-verify cycle. Finally, the number of accepted tokens $L_A$ in a cycle (termed acceptance length) is divided by the sum of draft time and verification time to calculate the throughput for this cycle. This throughput is used as the reward signal to optimize our depth policy and size policy.

## 4 METHOD

We frame the interaction between draft and target models within a Reinforcement Learning (RL) framework, employing two co-adaptive policies to navigate the inherent trade-off between acceptance length and latency: a **depth policy**, $\pi_D$, which determines the draft depth, and a **size policy**, $\pi_V$, which determines the verification size. The action space for the depth policy is binary, $\mathcal{A}_{\pi_D} = \{0, 1\}$, determining whether to continue drafting after each forward pass, while the action space for the size policy, $\mathcal{A}_{\pi_V} \in [20, 240]$, is a discrete range of verification sizes. To inform their decisions, the policies observe a state vector $s_t \in \mathcal{S}$ at each time step $t$, defined as

$$s_t = [D, L, P_1; \ldots; P_{V_{all}}] \tag{3}$$

where $D$ is the current draft depth, $L$ is the context length, and $P_1, \ldots, P_{V_{all}}$ are the predicted probabilities for all tokens in the draft tree. This state representation is designed to be highly informative, as token probabilities are strong predictors of acceptance length (Li et al., 2025a), while draft depth and sequence length are the important determinants of latency (Figure 2). To optimize decision latency, the size policy $\pi_V$ observes the entire state, whereas the depth policy $\pi_D$ utilizes a minimal subset: the probability of tokens at the last layer of the draft tree, along with $D$ and $L$. Both policies are implemented as lightweight Multi-Layer Perceptrons (MLPs) to ensure the decision-making process introduces negligible computational overhead. More details about the two policies can be found in Appendix A.2.

These policies are jointly optimized using the Proximal Policy Optimization (PPO) algorithm (Schulman et al., 2017) to maximize a reward signal, $R_t$, which represents the throughput for each cycle:

$$R_t = L_A / (T_{draft} + T_{verify}) \tag{4}$$

This reward function directly incentivizes the policies to master the complex trade-off between generating more potentially correct tokens and the time cost incurred. We update the policy network $\pi_\theta$ through the following objective function:

$$J(\theta) = \hat{\mathbb{E}}_t \left[ \min \left( \frac{\pi_\theta(a_t | s_t)}{\pi_{\theta_{old}}(a_t | s_t)} \hat{A}_t, \text{ clip} \left( \frac{\pi_\theta(a_t | s_t)}{\pi_{\theta_{old}}(a_t | s_t)}, 1 - \epsilon, 1 + \epsilon \right) \hat{A}_t \right) \right], \tag{5}$$

where $\pi_{\theta_{old}}$ is the behavioral policy and $\hat{A}_t$ is the advantage estimate (Schulman et al., 2017). Following each optimization step, the old policy parameters are updated with the new ($\theta_{old} \leftarrow \theta$),

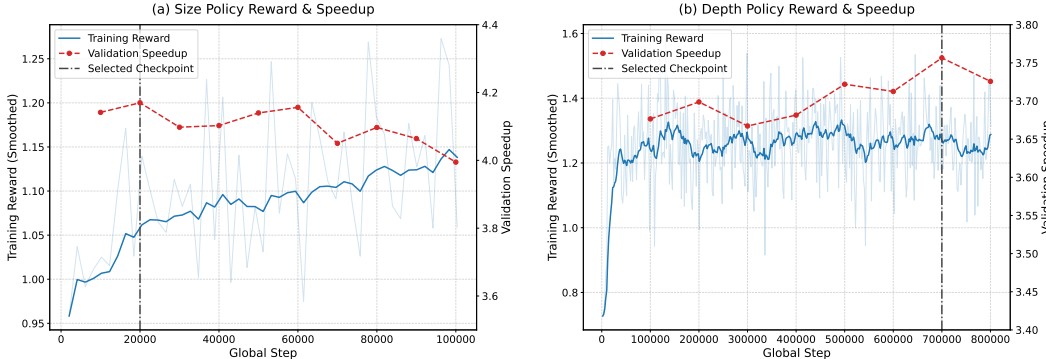

Figure 3: We report the training reward curves for the Size Policy and Depth Policy of the Vicuna-13B model, as well as the evolution of the speedup ratio on the validation set.

and the process of data collection and optimization is repeated. To ensure stable convergence, we employ a two-stage training procedure. The first stage involves training the initial depth and size policies independently. In the second stage, we use these initial policies as a starting point for iterative training, allowing them to co-adapt and converge on a coordinated strategy.

**Initial Policy Training.** We leverage the HumanEval dataset (Chen et al., 2021) for training, as its diverse and complex code snippets provide a rich environment for the policies to develop a wide range of adaptive strategies. This initial phase involves training each policy independently while its counterpart operates on a fixed, heuristic-based strategy. Specifically, the size policy, $\pi_V$, is trained for 100k steps with draft depths randomly sampled from the range $[1, 12]$. Conversely, the depth policy, $\pi_D$, is trained for 1M steps using a constant verification size of 60, mirroring the heuristic used in the Eagle3 baseline. This initial training establishes robust, standalone policies that serve as a stable foundation for the subsequent co-adaptation phase. We present the evolution of the reward for the Vicuna-13B policies during training in Figure 3, alongside the speedup ratios observed on the validation set. The checkpoint yielding the highest performance on the validation set was selected as the basis for subsequent training. As illustrated in the figure, the number of training steps was sufficient to ensure model convergence. The training curves of the other models exhibit patterns similar to those of Vicuna-13B.

**Iterative Optimizing for Policy Co-Adaptation.** Although the policies are trained independently, achieving optimal performance at inference hinges on their synergistic operation. To address this co-adaptation challenge, we introduce an iterative optimization procedure. This process alternates between refining each policy while holding the other fixed. First, we freeze the parameters of the size policy ($\pi_V$) and optimize the depth policy ($\pi_D$) with respect to the dynamic verification sizes $\pi_V$ proposes. Subsequently, the newly updated depth policy ($\pi_D$) is frozen, and the size policy ($\pi_V$) is optimized based on the dynamic draft depths determined by $\pi_D$. This alternating optimization cycle allows the two policies to progressively co-adapt to each other's evolving strategies. Empirically, we found that only two rounds of this iterative process yield a highly synergistic strategy and substantial performance gains.

## 5 EXPERIMENTS

### 5.1 SETUPS

**Models** We conduct experiments with state-of-the-art open source chat and reasoning models, including Llama-3.1-8B-Instruct (Dubey et al., 2024), Vicuna-13B-v1.3 (Chiang et al., 2023), DeepSeek-R1-Distill-LLaMA 8B (DeepSeek-AI, 2025), Qwen3-14B (Team, 2025a) and Qwen3-32B (Team, 2025a). The Eagle3 models for the Qwen3 series are from AngelSlim (Contributors, 2025), as the official implementation had not been released. For all other models, we used the official public releases of Eagle3 (Li et al., 2025a).

**Tasks**    Following EAGLE (Li et al., 2024) and Spec-Bench (Xia et al., 2024), we evaluate on four common tasks, using the same policy weights for all tasks. For multi-turn conversation, mathematical reasoning, instruction following, and QA, we chose the MT-bench (Zheng et al., 2023), GSM8K (Cobbe et al., 2021), Alpaca (Taori et al., 2023) and Natural Questions (Kwiatkowski et al., 2019) datasets, respectively.

**Metrics**    Speculative decoding will not change the output distribution for target model. Therefore,we do not evaluate the generation quality and instead use the following common used metrics (Li et al., 2025a) to assess acceleration performance:

- **Speedup:** The actual test speedup ratio relative to vanilla auto-regressive decoding.
- **Average Acceptance Length $\tau$:** The average number of tokens generated per draft-and-verify cycle on the benchmark dataset.

**Implementation Details**    We implement the Proximal Policy Optimization (PPO) algorithm using the Stable-Baselines3 library (Raffin et al., 2021) and train our policies on the HumanEval dataset (Chen et al., 2021). The size and depth policies are trained for 100k and 1M steps, respectively. Further analysis regarding the choice of training steps can be found in the Appendix C.1. The PPO is configured with a rollout buffer of 2048 steps and a minibatch size of 256. Due to the lightweight nature of the two policies, the RL training phase is highly efficient, detailed can be found in Appendix A.7. A comprehensive list of all hyperparameters can be found in Appendix A.3.

**Comparison**    We use vanilla auto-regressive decoding as the baseline, which serves as the benchmark for speedup ratios (1.00x). We apply our methods to current SOTA Speculative Decoding method Eagle3 and compare with recent representative dynamic depth methods, including DDD (Brown et al., 2024), SVIP (Zhang et al., 2024c), Gammatune (Gautam et al., 2025), Disco (Mamou et al., 2024) and SpecDec++ (Huang et al., 2025a) and dynamic size methods C2T (Huo et al., 2025). We set the default parameters for dynamic depth methods with a verification size 60 and a maximum depth of 12, dynamic size methods with a depth of 8. For the Eagle3 baseline, we set the verification size to 60, 50 for 8B, 13B original LLMs respectively, with a draft depth of 8, and select 10 nodes during the expansion phase as provided by the original implements (Li et al., 2025a). We also employed Grid Search(GS) to find the optimal hyper-parameters for Eagle3, as a strong baseline. More details can be seen in Appendix A.1.

## 5.2 EFFECTIVENESS

The performance of our method is detailed in Table 1 and Table 5. The experimental results demonstrate that our proposed method consistently outperforms all competing approaches across the evaluated models, datasets, and sampling temperatures. The stable speedup observed across diverse configurations highlights the strong capability of our framework.

Our method accelerates inference by an average of 6.5% on Llama-3 and 5.1% on Vicuna compared to the default Eagle3 configuration. In long thinking model like Deepseek-R1-Distll-LLaMA 8B, our method can yield 10% accelerate, which is quite useful as the long generation time for these models. For Qwen3 series, as no default parameters were provided, we report the results of grid search as our baseline. Our method shows significantly greater improvement on the 32B model compared to the 14B model. This demonstrates the reliability of our method at larger scales. Notably, our method maintains the highest average speedup at a temperature of 1.0 as shown in Table 5, despite being trained exclusively at greedy decoding. In contrast, other dynamic methods that adjust draft depth or verification size, while effective at greedy decoding, exhibit significant performance degradation at the higher temperature. This highlights the strong robustness of our method.

Our method does not necessarily yield the longest acceptance length; its $\tau$ is often shorter than that of the Grid Search-optimized baseline. Nevertheless, it achieves the highest overall speedup. This outcome reveals the core mechanism of our approach: the reinforcement learning framework makes policies learn to optimize directly for decoding efficiency. It strategically forgoes candidate tokens that, while potentially correct, would incur a disproportionately high time cost. By directly optimizing for the reward signal of throughput, our method achieves a superior speedup ratio by adaptively balancing acceptance length against time cost.

Table 1: Speedup ratios and average acceptance lengths $\tau$ of different methods during greedy decoding. *L31 8B* represents Llama-3.1-8B-Instruct, *V 13B* represents Vicuna-13B-v1.3, *Dpsk 8B* represents DeepSeek-R1-Distill-LLaMA 8B. For the Qwen3-14B and Qwen3-32B models, their corresponding Eagle3 models lack default settings. Therefore, only the results from Grid Search are presented. GT, Spec++, and Gs stand for Gammatune, SpecDec++, and Grid Search, respectively. The results of Temperature=1 are presented in Table 5.

| Model | Method | MT-bench | | Gsm8k | | Alpaca | | Qa | | Mean | |
|---|---|---|---|---|---|---|---|---|---|---|---|
| | | Speedup | $\tau$ | Speedup | $\tau$ | Speedup | $\tau$ | Speedup | $\tau$ | Speedup | $\tau$ |
| | | Temperature=0 | | | | | | | | | |
| | Eagle3 | 3.70 | 6.39 | 3.58 | 6.44 | 4.26 | 6.93 | 3.17 | 5.35 | 3.68 | 6.28 |
| | Eagle3+DDD | 3.67 | 5.41 | 3.55 | 5.48 | 3.97 | 5.58 | 3.07 | 4.21 | 3.57 | 5.17 |
| | Eagle3+Svip | 3.65 | 6.21 | 3.47 | 6.21 | 4.04 | 6.73 | 3.12 | 4.78 | 3.57 | 5.98 |
| | Eagle3+GT | 3.79 | 5.86 | 3.64 | 5.77 | 4.37 | 6.49 | 3.38 | 4.92 | 3.80 | 5.76 |
| *L31 8B* | Eagle3+C2T | 3.80 | 6.39 | 3.57 | 6.44 | 4.19 | 6.93 | 3.25 | 5.36 | 3.70 | 6.28 |
| | Eagle3+Disco | 3.70 | 6.11 | 3.56 | 6.18 | 4.14 | 6.51 | 3.22 | 5.05 | 3.66 | 5.96 |
| | Eagle3+Spec++ | 3.82 | 6.09 | 3.65 | 6.08 | 4.29 | 6.70 | 3.28 | 5.23 | 3.76 | 6.03 |
| | Eagle3+GS | 3.83 | **6.75** | 3.78 | **6.91** | 4.38 | **7.42** | 3.33 | **5.67** | 3.83 | **6.69** |
| | Eagle3+LTD | **3.96** | 6.74 | **3.79** | 6.73 | **4.53** | 7.31 | **3.41** | 5.44 | **3.92** | 6.56 |
| | Eagle3 | 4.48 | 7.22 | 4.36 | 6.58 | 4.16 | 6.47 | 3.43 | 5.12 | 4.11 | 6.35 |
| | Eagle3+DDD | 4.15 | 6.69 | 4.25 | 6.22 | 3.98 | 5.50 | 3.22 | 4.34 | 3.90 | 5.69 |
| | Eagle3+Svip | 4.02 | 7.74 | 3.86 | 6.79 | 3.77 | 6.53 | 3.02 | 4.96 | 3.67 | 6.51 |
| | Eagle3+GT | 4.15 | 6.85 | 4.07 | 6.00 | 3.94 | 6.09 | 3.35 | 4.81 | 3.88 | 5.94 |
| *V 13B* | Eagle3+C2T | 4.33 | 7.22 | 3.97 | 6.58 | 3.93 | 6.47 | 3.23 | 5.12 | 3.87 | 6.35 |
| | Eagle3+Disco | 4.48 | 7.08 | 4.13 | 6.49 | 4.02 | 6.29 | 3.31 | 5.05 | 3.99 | 6.23 |
| | Eagle3+Spec++ | 4.25 | 6.78 | 4.09 | 6.16 | 4.00 | 6.27 | 3.21 | 5.03 | 3.89 | 6.06 |
| | Eagle3+GS | 4.53 | 7.99 | 4.47 | 7.27 | 4.38 | **7.16** | 3.51 | **5.48** | 4.22 | 6.98 |
| | Eagle3+LTD | **4.64** | **8.24** | **4.59** | **7.38** | **4.40** | 7.01 | **3.66** | 5.37 | **4.32** | **7.00** |
| | Eagle3 | 3.67 | 5.82 | 4.73 | 7.39 | 3.65 | 5.63 | 3.14 | 5.02 | 3.80 | 5.53 |
| *Dpsk 8B* | Eagle3+GS | 3.69 | **6.42** | 4.83 | 8.37 | 3.54 | **6.24** | 3.20 | **5.55** | 3.82 | 6.08 |
| | Eagle3+LTD | **4.05** | 6.32 | **4.98** | **8.53** | **3.98** | 6.00 | **3.62** | 5.37 | **4.16** | **6.08** |
| *Qwen3 14B* | Eagle3+GS | 2.34 | **3.40** | 2.75 | **3.95** | 2.15 | **2.90** | 1.86 | **2.65** | 2.28 | **3.23** |
| | Eagle3+LTD | **2.44** | 3.34 | **2.82** | 3.92 | **2.24** | 2.87 | **1.98** | 2.62 | **2.37** | 3.19 |
| *Qwen3 32B* | Eagle3+GS | 1.76 | **3.30** | 2.26 | **4.10** | 1.75 | **2.96** | 1.45 | **2.67** | 1.81 | **3.26** |
| | Eagle3+LTD | **2.42** | 3.17 | **3.02** | 3.90 | **2.38** | 2.85 | **2.06** | 2.60 | **2.47** | 3.13 |

To better verify generalization, we evaluated our method on MMLU, which consists of 57 subtasks across diverse domains (results detailed in Appendix B and Figure 7). Our method outperforms Eagle3 on 54 out of 57 subtasks, achieving an average speedup improvement of 5%. The top-10 performing categories vary widely, including not only Math, Logic, but also Human Aging, and Government Politics. These results strongly support the cross-domain generalization of our approach.

## 5.3 Draft Time and Verification Time

To provide a granular analysis of our performance gains, we present the relative composition of the total inference time, breaking it down into its draft and verification components in Figure 2 Right. It reveals that our method's speedup stems from improvements in both drafting and verification. Universally, draft time is reduced due to our dynamic depth policy, which curtails speculation in low-confidence scenarios to save computation. Notably, our method can also reduce the verification time, implying a lower token verification load. Paradoxically, this lower overhead corresponds to a higher average acceptance length for Llama,Vicuna and Dpsk models as shown in Table 1. This outcome highlights that our method's strength lies not just in reducing the size of draft trees, but in strategically allocating candidate tokens to contexts where they are most likely to be accepted, which validate the efficacy of our approach. For the Qwen series models, while our method exhibits a lower acceptance length compared to the baselines, it significantly reduces verification time, especially for the Qwen3-32B model, thereby achieving a substantially higher speedup ratio.

## 6 Ablation Study

In this section, we aim to validate the effectiveness of LTD method through a series of analyses. We begin in Section 6.1, demonstrate the effectiveness of the iterative optimizing process. Then we

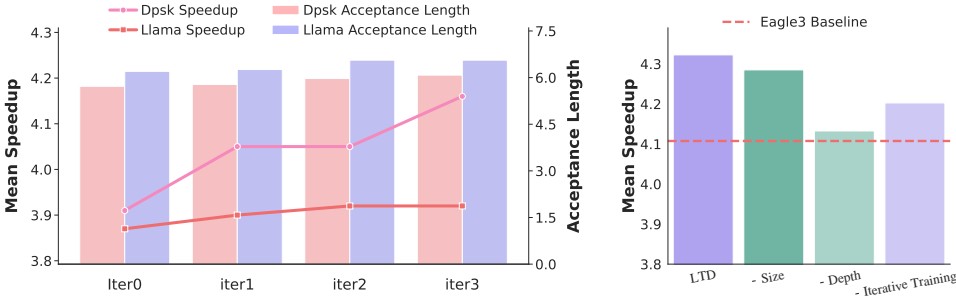

Figure 4: Left: The effectiveness of our iterative training strategy on Llama3-8B and Dpsk-8B models. Right: The contribution of each component on Vicuna-13B model.

conduct an ablation study that isolates the contributions of the depth and size policies in Section 6.2. To justify our reinforcement learning design, we further experiment with other reward signals in Section 6.3 and observation spaces in Appendix A.4. Taken together, these experiments underscore the effectiveness of each component within our LTD method.

## 6.1 ANALYSIS OF ITERATIVE OPTIMIZING

We investigated the effectiveness of our iterative optimization strategy on the Llama3-8B and Dpsk-8B models. The results are presented in Figure 4 (Left), with comprehensive data available in Table 7 and Table 8 in Appendix A.5. The process begins at Iteration 0 (Iter0) with a naive combination of the initially trained policies. Subsequently, we apply the alternating optimization schedule described in Section 4. Specifically, we first optimize the size policy (Iter1), then the depth policy (Iter2), and finally the size policy again (Iter3). During each stage, the parameters of the other policy are held frozen. The results affirm the benefits of this iterative approach.

Each optimization step yields progressive improvements in both mean speedup and acceptance length, confirming that the policy co-adaptation leads to a more synergistic system. While the most significant gains occur in the initial transition from Iter0 to Iter1, the impact of subsequent iterations is model-dependent. For Llama3-8B, performance saturates after Iter2, with Iter3 offering only negligible benefits. In contrast, the Dpsk-8B model continues to show substantial improvement through Iter3. These findings validate the effectiveness of our iterative optimizing strategy.

## 6.2 IMPACT OF KEY COMPONENTS

To analyze the contribution of key components in our proposed method, we conducted a comprehensive ablation study on the Vicuna-13B model presented in Figure 4 (Right). The detailed results are presented in Table 4 in Appendix A.5. We evaluated three primary variants: 1) −Size: The size policy was ablated, relying solely on the depth policy. 2) −Depth: The depth policy was ablated, relying solely on the size policy. 3) −Iterative Training: The initial token and depth policies from Section 4 were directly combined without our proposed iterative optimizing.

The results clearly demonstrate that the depth policy plays an important role to the overall acceleration. While the size policy also yields a speedup over the baseline, its impact is less pronounced. Furthermore, a naive combination of the two policies leads to suboptimal performance. This highlights the efficacy of our iterative training scheme, which allows the policies to co-adapt and achieve a synergistic acceleration. In summary, these results validate the individual contributions of both policies and the essential role of the co-adaptive process that unites them.

## 6.3 EFFECTIVENESS OF THE REWARD SIGNAL

To validate the effectiveness of using throughput as the reward signal, we designed and compared three distinct reward functions to optimize our size policy alone on Vicuna model: 1) the acceptance length per cycle, 2) the time cost per cycle, and 3) throughput per cycle. Our results in Table 2 demonstrate that employing throughput as the reward yields the highest speedup across all datasets.

Table 2: The effectiveness of reward signal on size policy, evaluated on Vicuna-13B model.

| Reward | MT-bench | | Gsm8k | | Alpaca | | Qa | | Mean | |
|---|---|---|---|---|---|---|---|---|---|---|
| | Speedup | $\tau$ | Speedup | $\tau$ | Speedup | $\tau$ | Speedup | $\tau$ | Speedup | $\tau$ |
| Acceptance Length | 3.77 | **7.70** | 3.81 | **7.20** | 3.71 | **7.05** | 3.06 | **5.61** | 3.59 | **6.89** |
| Time Cost | 3.89 | 6.49 | 3.54 | 5.80 | 3.48 | 5.79 | 2.87 | 4.54 | 3.45 | 5.66 |
| Throughput | **4.40** | 7.68 | **4.43** | 7.10 | **4.23** | 7.00 | **3.47** | 5.56 | **4.13** | 6.84 |

While using acceptance length as the reward achieves the maximum $\tau$, its average speedup is substantially lower. This discrepancy occurs because this size policy proposes an excessive number of tokens for verification, leading to a significant increase in verification time that outweighs the benefits of a higher acceptance length. Conversely, using time cost as the reward leads to a lowest acceptance length, as the policy is encouraged to propose too few tokens, which cannot benefit form the parallel verification from the target model, ultimately resulting in the lowest overall speedup. These findings confirm that throughput is a superior reward signal as it inherently balances the competing objectives of maximizing acceptance length and minimizing time cost.

## 7 Conclusion

In this paper, we introduce LTD, a novel speculative decoding method that directly optimizes the throughput of each draft-and-verify cycle using reinforcement learning. By employing two co-adaptive policies to dynamically coordinate the drafting and verification phases, LTD learns to manage the trade-off between acceptance length and time cost, achieving a synergistic acceleration. Extensive experiments on five distinct LLMs across four benchmarks demonstrate that LTD consistently achieves state-of-the-art acceleration for both greedy decoding and high-temperature sampling. This superior performance underscores the effectiveness and robustness of our approach, positioning LTD as a powerful solution for accelerating LLM inference.

## Reproducibility Statement

In Section 4 and Appendix A.2, we provide a detailed description of our method and the training process; in Section 5.1 and Appendix A.3, we present the complete training parameters and evaluation datasets; and in Appendix A.1, we document the hyperparameters used for baseline methods. Together, these details support the reproducibility of our work.

## Acknowledgement

We thank the anonymous reviewers for their helpful comments on this paper. This work was partially supported by National Natural Science Foundation of China projects (No. 92470205 and 62476010).

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

# A  DETAILED SETTINGS

## A.1  BASELINES SETTINGS

Table 3 summarizes the key difference between our methods and various dynamic draft tree methods. For DDD, cumulative log-probabilities were monitored after the 5th, 7th, and 9th layers, with a minimum threshold of $-0.3$. In Gammatune, $\gamma_{\min}$ was half the default depth, $\gamma_{\max}$ was $1.5$ times the default depth, and the adaptation rate $\eta$ was $0.2$. For SVIP, we followed the original paper, using an entropy factor of $0.15$ and a minimum entropy of $0.3$. For DDD and SVIP, we directly adopted the hyperparameters provided by the original authors, while we determined the hyperparameters ourselves based on experiments as the original paper did not specify them.

For FFN-based methods, samples from HumanEval generated by default Eagle3 were used to extract hidden states, token scores, entropy, and a binary flag indicating whether a draft token was accepted by the target model. Equal numbers of positive and negative samples were used for training. C2T was trained with a hidden size of $64$ for 50 epochs, while Disco used a hidden size of $128$ with the same training epochs; SpecDec++ employed a ResNet with hidden size $1024$ for 5 epochs. All models were trained with AdamW (lr $= 1e{-}4$). Interestingly, we observed that including entropy in the observation space can sometimes reduce speedup. This is due to a tradeoff between the overhead of frequent entropy computation and the accuracy of the FFN decision. Based on our validation experiments, we used entropy as an observation feature for Disco, while omitting it for C2T.

We also performed a grid search to determine the optimal hyperparameters for the Eagle3 model on the HumanEval benchmark as a strong baseline. Our search space included draft depths ranging from 4 to 12 and verification sizes from 40 to 240. The configuration that yielded the best performance was then selected for subsequent experiments.

## A.2  LTD

### A.2.1  SIZE POLICY

The Size Policy governs the selection of the verification size, $V$—the number of candidate tokens from the speculative tree that are passed to the target model. Its objective is to assess the aggregate quality of the draft tree and select a value for $V$ that optimally balances the computational cost of verification against the potential number of accepted tokens.

- **Architecture:** Both the policy and value networks are implemented as two-layer feedforward networks (FFNs) with hidden layer dimensions of [1024, 256].
- **Training:** To promote robustness to varying draft tree structures, the policy is trained on episodes where the generation depth is randomized, sampled uniformly from integers between 1 and 12.
- **Observation Space:** The state representation provided to the agent is designed to include the key outcome of the drafting phase. It is a fixed-size vector primarily composed of the log-probability scores of all candidate tokens from the generated tree. For a tree of depth $D$, this includes scores for up to $1 + W + (D - 2) \times W^2$ tokens, with padding applied to ensure a consistent vector length. This score vector is then concatenated with scalar features representing the final draft depth ($D$) and the total length of the input sequence($L$).
- **Action Space:** The agent's action space is discrete, comprising 12 actions that map linearly to a verification size $V$ ranging from 20 to 240.

### A.2.2  DEPTH POLICY

The Depth Policy executes a sequential policy, deciding at each depth level whether to continue expanding the draft tree (`CONTINUE`) or to terminate the drafting phase and proceed to verification (`STOP`). Given that this policy is executed multiple times within a single generation step, its inference latency is a critical performance factor.

- **Architecture:** To minimize this latency, we utilize a lightweight architecture. We find that a single-layer FFN(with hidden layer dimensions 1024) achieves an effective balance between predictive performance and inference speed.

Table 3: Comparison of Different Dynamic Draft Tree Methods

| Method Name | Predict Policy | Observation Space | Dynamic Size or Depth | Training Methods | Objective |
|---|---|---|---|---|---|
| DDD (Brown et al., 2024) | Heuristic | Probability | Depth | - | $L_A$ |
| SVIP (Zhang et al., 2024c) | Heuristic | Entropy | Depth | - | $L_A$ |
| Gammatune (Gautam et al., 2025) | Heuristic | Token Acceptance Rate | Depth | - | $L_A$ |
| Disco (Mamou et al., 2024) | FFN | Probability, Entropy, Draft Depth | Depth | FT | $L_A$ |
| SpecDec++ (Huang et al., 2025a) | FFN | Hidden States | Depth | FT | $L_A$ |
| C2T (Huo et al., 2025) | FFN | Probability, Entropy, Draft Depth | Size | FT | $L_A$ |
| LTD | FFN | Probability, Total Length, Draft Depth | Both | RL | Throughput |

- **Observation Space:** The state for this sequential decision-making process is updated at each depth level. The observation vector is primarily composed of the log-probabilities of the candidate tokens generated at the current expansion frontier. For a beam search of width $W$, this comprises $W^2$ candidate scores. This probability vector is then concatenated with two scalar features: the current draft depth ($D$) and the total length of the input sequence($L$).

- **Training Strategy:** To establish a stable training curriculum, we first trained an initial depth policy against a fixed verification size of $V = 60$ tokens, which mirrors the heuristic used in the Eagle3 baseline.

- **Action and Reward:** The agent outputs a binary action at each step. A numerical reward, based on the final throughput of the episode, is dispensed only upon selecting the STOP action. Intermediate CONTINUE actions yield zero reward. To counteract the policy's potential bias towards premature termination, we employ a high discount factor ($\gamma = 0.999$), thereby encouraging the exploration of deeper draft trees to get more rewards potentially.

Table 4: Contribution of each components of LTD, evaluated on Vicuna-13B.

| Method | MT-bench | | Gsm8k | | Alpaca | | QA | | Mean | |
|---|---|---|---|---|---|---|---|---|---|---|
| | Speedup | $\tau$ | Speedup | $\tau$ | Speedup | $\tau$ | Speedup | $\tau$ | Speedup | $\tau$ |
| Eagle3 | 4.48 | 7.22 | 4.36 | 6.58 | 4.16 | 6.47 | 3.43 | 5.12 | 4.11 | 6.35 |
| -Depth | 4.40 | 7.68 | 4.43 | 7.10 | 4.23 | 7.00 | 3.47 | **5.56** | 4.13 | 6.84 |
| -Size | **4.77** | 7.75 | 4.46 | 6.82 | 4.36 | 6.59 | 3.55 | 5.12 | 4.29 | 6.57 |
| -Iterative Training | 4.54 | **8.29** | 4.48 | **7.41** | 4.28 | **7.08** | 3.51 | 5.45 | 4.20 | **7.06** |
| LTD | 4.64 | 8.24 | **4.59** | 7.38 | **4.40** | 7.01 | **3.66** | 5.37 | **4.32** | 7.00 |

## A.3 TRAINING DETAILS FOR LTD

The policy (actor) and value (critic) functions are both implemented as Multi-Layer Perceptrons (MLPs).

**Size Policy ($\pi_V$)**   The policy and value networks for the size part share an architecture consisting of two hidden layers with 1024 and 256 units respectively, using ReLU activation functions. This policy is trained with a standard discount factor of $\gamma = 0.9$.

**Depth Policy ($\pi_D$)**   To minimize decision latency, the depth policy utilizes a more lightweight architecture with a single hidden layer of 1024 units, while the value networks remains same to the size part. A key distinction is the use of a high discount factor ($\gamma = 0.999$). This value is chosen specifically to mitigate the policy's potential bias towards premature termination, thereby encouraging the exploration of deeper draft trees by prioritizing long-term rewards.

**Shared Parameters**   For both policies, the PPO surrogate loss is optimized for 20 epochs per update cycle, and an entropy coefficient of 0.01 is applied to foster exploration. We use a learning rate of 1e-3, which decays linearly to zero following a warm-up phase over the initial 1% of training steps. All other hyperparameters were kept at their default values as provided by the Stable-Baselines3 library.

Table 5: Speedup ratios and average acceptance lengths $\tau$ of different methods during Temperature=1. *L31 8B* represents Llama-3.1-8B-Instruct, *V 13B* represents Vicuna-13B-v1.3, *Dpsk 8B* represents DeepSeek-R1-Distill-LLaMA 8B. For the Qwen3-14B and Qwen3-32B models, their corresponding Eagle3 models lack default settings. Therefore, only the results from Grid Search are presented. GT, Spec++, and Gs stand for Gammatune, SpecDec++, and Grid Search, respectively.

| Model | Method | MT-bench | | Gsm8k | | Alpaca | | Qa | | Mean | |
|---|---|---|---|---|---|---|---|---|---|---|---|
| | | Speedup | $\tau$ | Speedup | $\tau$ | Speedup | $\tau$ | Speedup | $\tau$ | Speedup | $\tau$ |
| | | | | | Temperature=1 | | | | | | |
| | Eagle3 | 2.85 | 4.11 | **3.43** | 5.24 | 3.38 | 5.47 | 2.27 | 3.06 | 2.98 | 4.47 |
| | Eagle3+DDD | **2.89** | 3.40 | 3.35 | 4.09 | 3.38 | 4.59 | 2.36 | 2.95 | 3.00 | 3.76 |
| | Eagle3+Svip | 2.74 | 3.41 | 3.21 | 4.92 | 3.19 | 4.95 | 2.31 | 3.20 | 2.86 | 4.12 |
| | Eagle3+GT | 2.75 | 3.54 | 3.28 | 4.27 | **3.51** | 5.12 | **2.46** | 3.24 | 3.00 | 4.04 |
| L31 8B | Eagle3+C2T | 2.78 | **4.11** | 3.28 | 5.24 | 3.24 | 5.47 | 2.27 | 3.06 | 2.89 | 4.47 |
| | Eagle3+Disco | 2.82 | 3.82 | 3.26 | 5.08 | 3.30 | 5.18 | 2.20 | 2.88 | 2.90 | 4.24 |
| | Eagle3+Spec++ | 2.92 | 4.20 | 3.33 | 4.74 | 3.42 | 5.51 | 2.35 | 3.28 | 3.01 | 4.43 |
| | Eagle3+GS | 2.79 | 4.08 | 3.36 | **5.50** | 3.46 | **5.79** | 2.31 | **3.40** | 2.98 | **4.69** |
| | Eagle3+LTD | 2.85 | 4.02 | 3.35 | 5.09 | 3.43 | 5.68 | 2.39 | 3.26 | **3.01** | 4.51 |
| | Eagle3 | 3.60 | 5.55 | 3.68 | 5.64 | 3.40 | 5.49 | 3.08 | 4.64 | 3.44 | 5.33 |
| | Eagle3+DDD | 3.40 | 5.05 | 3.58 | 5.46 | 3.43 | 5.00 | 2.96 | 4.11 | 3.34 | 4.91 |
| V 13B | Eagle3+Svip | 3.18 | 5.73 | 3.26 | 5.78 | 3.21 | 5.39 | 2.61 | 4.44 | 3.07 | 5.34 |
| | Eagle3+GT | 3.43 | 5.36 | 3.32 | 5.26 | 3.39 | 4.95 | 2.99 | 4.44 | 3.28 | 5.00 |
| | Eagle3+C2T | 3.38 | 5.66 | 3.41 | 5.63 | 3.27 | 5.54 | 2.83 | 4.65 | 3.22 | 5.37 |
| | Eagle3+Disco | 3.48 | 5.55 | 3.67 | 5.74 | 3.36 | 5.51 | 2.88 | 4.59 | 3.35 | 5.35 |
| | Eagle3+Spec++ | 3.61 | 5.58 | 3.64 | 5.63 | 3.34 | 5.39 | 3.02 | 4.55 | 3.40 | 5.29 |
| | Eagle3+GS | 3.58 | 6.30 | **3.81** | **6.39** | 3.61 | 6.01 | 3.00 | 4.93 | 3.50 | 5.91 |
| | Eagle3+LTD | **3.82** | **6.40** | 3.75 | 6.38 | **3.65** | **6.20** | **3.26** | 4.97 | **3.62** | **5.99** |
| | Eagle3 | 3.07 | 4.74 | 3.97 | 6.89 | 2.79 | 4.53 | 2.44 | 4.06 | 3.07 | 4.66 |
| Dpsk 8B | Eagle3+GS | 3.05 | 5.19 | **4.14** | 7.64 | 2.76 | **4.83** | 2.33 | **4.44** | 3.07 | 5.03 |
| | Eagle3+LTD | **3.18** | **5.20** | 4.02 | **7.95** | **2.83** | 4.76 | **2.57** | 4.39 | **3.15** | **5.09** |
| Qwen3 14B | Eagle3+GS | 2.05 | **3.13** | 2.62 | **3.83** | 2.04 | **2.78** | 1.68 | **2.57** | 2.10 | **3.08** |
| | Eagle3+LTD | **2.16** | 3.11 | **2.73** | 3.85 | **2.11** | 2.76 | **1.82** | 2.55 | **2.21** | 3.07 |
| Qwen3 32B | Eagle3+GS | 1.56 | **2.89** | 2.10 | **3.95** | 1.63 | **2.73** | 1.33 | **2.46** | 1.66 | **3.01** |
| | Eagle3+LTD | **2.17** | 2.78 | **2.83** | 3.76 | **2.16** | 2.59 | **1.88** | 2.41 | **2.26** | 2.89 |

Table 6: Ablation Study for Observation Space on size policy evaluated on Vicuna-13B.

| Reward | MT-bench | | Gsm8k | | Alpaca | | Qa | | Mean | |
|---|---|---|---|---|---|---|---|---|---|---|
| | Speedup | $\tau$ | Speedup | $\tau$ | Speedup | $\tau$ | Speedup | $\tau$ | Speedup | $\tau$ |
| P+L | 4.35 | 7.68 | 4.30 | 7.10 | 4.18 | 7.00 | 3.35 | 5.55 | 4.05 | 6.83 |
| P+D | 4.31 | 7.68 | 4.35 | 7.10 | 4.22 | 7.00 | 3.41 | 5.56 | 4.07 | 6.84 |
| P+D+L (LTD) | **4.40** | 7.68 | **4.43** | 7.10 | 4.23 | 7.00 | **3.47** | 5.56 | **4.13** | 6.84 |
| P+D+L+H | 4.40 | 7.53 | 4.32 | 6.97 | 4.22 | 6.94 | 3.44 | 5.51 | 4.10 | 6.74 |
| P+D+L+E | 4.36 | 7.68 | 4.36 | 7.10 | **4.28** | 7.00 | 3.43 | 5.56 | 4.11 | 6.84 |

## A.4 Ablation Study For Observation Space

We conducted an ablation study to determine the optimal composition of the observation state for our policy. While prior methods have incorporated variables such as predicted probabilities (P), hidden states (H), entropy (E), input length (L), and draft depth (D) into their state representation, the inclusion of such computationally expensive features is only justified if the resulting accuracy gains outweigh the associated latency cost.

Our results in Table 6 indicate that our proposed observation space configuration achieves the highest average speedup across these four benchmarks. Notably, we observe that many competing methods yield similar acceptance lengths. This suggests that while their proposed verification sizes are sufficient to capture the ground-truth token sequence, this often comes at the cost of excessive verification time. The superiority of our method stems from two key advantages. First, compared to approaches that rely on computationally intensive inputs like entropy or full hidden states, our method incurs significantly lower overhead when constructing its observation state. Second, in contrast to baselines that omit draft depth or total length information, our model learns to propose a more

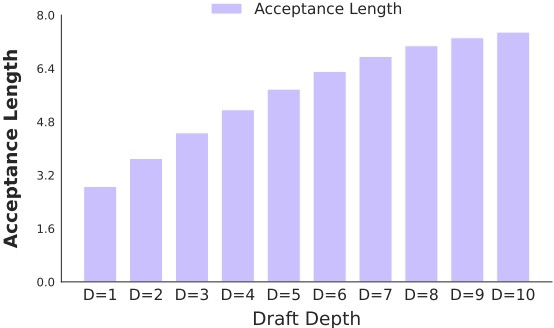 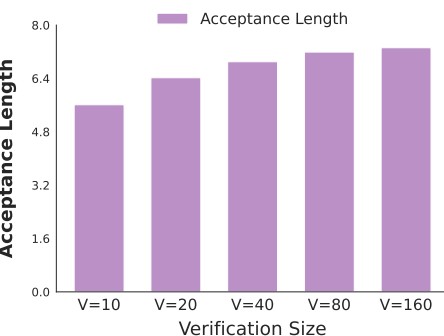

Figure 5: The average acceptance length for Llama-3.1-8B-Instruct using Eagle3 on HumanEval dataset. Left: Change the draft depth with verification size fixed to 60. Right: Change the verification size with draft depth fixed to 8.

conservative number of candidate tokens. This cautious strategy is crucial for reducing verification latency.

## A.5   DETAILED RESULTS FOR ABLATION STUDIES

The detailed results for ablation studies are shown in the section. It contains the results for contribution of each components of LTD in Table 4, and the results for iterative optimizing in Table 7 and Table 8.

Table 7: Llama3.1-8B Iterative Optimizing

| Method | MT-bench | | Gsm8k | | Alpaca | | Qa | | Mean | |
| | Speedup | $\tau$ | Speedup | $\tau$ | Speedup | $\tau$ | Speedup | $\tau$ | Speedup | $\tau$ |
| --- | --- | --- | --- | --- | --- | --- | --- | --- | --- | --- |
| Eagle3 | 3.70 | 6.39 | 3.58 | 6.44 | 4.26 | 6.93 | 3.17 | 5.35 | 3.68 | 6.28 |
| Iter0 | 3.88 | 6.46 | 3.76 | 6.40 | 4.34 | 6.82 | 3.50 | 5.12 | 3.87 | 6.20 |
| Iter1 | 3.95 | 6.50 | **3.80** | 6.47 | 4.36 | 6.89 | **3.50** | 5.18 | 3.90 | 6.26 |
| Iter2 | **3.96** | 6.74 | 3.79 | 6.73 | 4.53 | 7.31 | 3.41 | 5.44 | 3.92 | 6.56 |
| Iter3 | 3.93 | **6.74** | 3.73 | **6.73** | **4.57** | **7.31** | 3.44 | **5.44** | 3.92 | **6.56** |

Table 8: Dpsk-8B Iterative Optimizing

| Method | MT-bench | | Gsm8k | | Alpaca | | Qa | | Mean | |
| | Speedup | $\tau$ | Speedup | $\tau$ | Speedup | $\tau$ | Speedup | $\tau$ | Speedup | $\tau$ |
| --- | --- | --- | --- | --- | --- | --- | --- | --- | --- | --- |
| Eagle3 | 3.67 | 5.82 | 4.73 | 7.39 | 3.65 | 5.63 | 3.14 | 5.02 | 3.80 | 5.53 |
| Iter0 | 3.88 | 5.95 | 4.69 | 8.03 | 3.68 | 5.63 | 3.37 | 5.07 | 3.91 | 5.72 |
| Iter1 | 3.96 | 5.98 | 4.92 | 8.14 | 3.86 | 5.67 | 3.46 | 5.08 | 4.05 | 5.78 |
| Iter2 | 4.02 | 6.22 | 4.94 | 8.36 | 3.71 | 5.91 | 3.52 | 5.29 | 4.05 | 5.97 |
| Iter3 | **4.05** | **6.32** | **4.98** | **8.53** | **3.98** | **6.00** | **3.62** | **5.37** | **4.16** | **6.08** |

## A.6   INTERACTION BETWEEN THE DEPTH AND SIZE POLICIES

To validate the effectiveness of our iterative training, we analyzed the test set distributions of the initial separately trained policies ('Iter0') and the final iteratively trained LTD policies, as shown in Figure 6. Specifically, we examined the relationship between the number of candidate tokens and the draft depth conditioned on the acceptance length. As observed, Iter0 exhibits a static pattern regardless of acceptance length, predominantly clustering in the region where the draft depth is $< 6$ and the token count is $< 60$. Instances with a shallow depth ($< 6$) but a large token budget ($> 60$) are negligible. In contrast, LTD demonstrates superior adaptability across different acceptance lengths. For large acceptance lengths, a significant proportion ($43\%$) of cases utilize both a large token budget and a deeper tree (both $> 60$). Furthermore, when the acceptance length is small—indicating higher

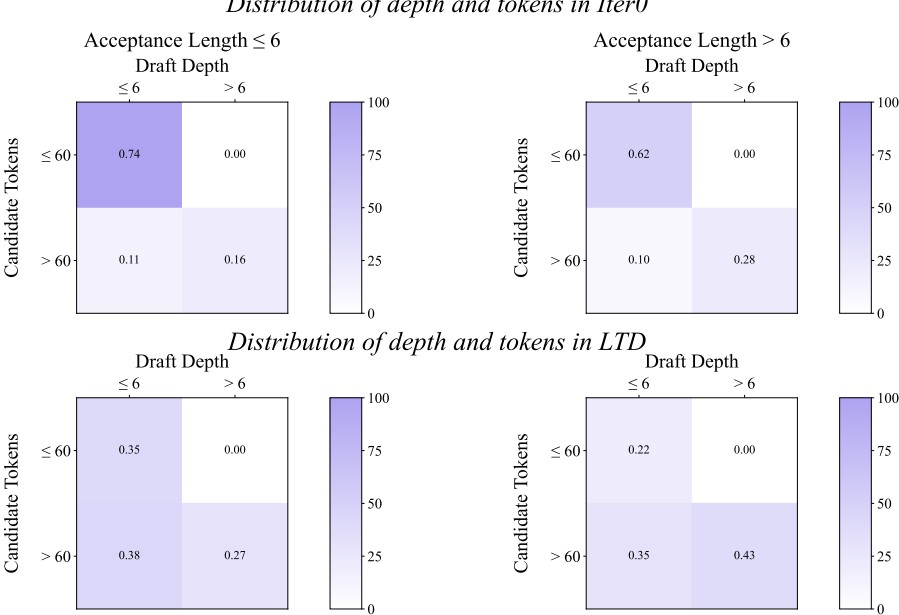

Figure 6: We report the interaction between depth policy and size policy on Llama3-8B for Iter0 and LTD.

generation difficulty—LTD frequently adopts a 'shallow but wide' strategy (depth $< 6$, tokens $> 60$). This strategy of deploying more candidate tokens effectively mitigates the reduction in acceptance length caused by increased difficulty.

### A.7 LATENCY OF DEPTH AND SIZE POLICIES

We analyzed the latency overhead of our depth and size policies on both Llama-3.1-8B-Instruct and Vicuna-13B-v1.3. Table 9 reports the average generation time per sample, along with the proportional time costs of the size and depth policies. The results demonstrate that the computational overhead introduced by our policies is minimal: the total overhead is less than 1.5% on Llama and even lower (1.2%) on the larger Vicuna model. Notably, the latency of the depth policy is approximately twice that of the size policy. Although we utilize a lightweight single-layer network, the depth policy incurs a higher total cost due to its more frequent invocation compared to the size policy.

### A.8 SIZE SELECTION FOR MLP POLICIES

There is an inherent trade-off in MLP design: we aim for the accuracy gains from a larger MLP to outweigh the increased inference cost. To validate our design choice, we conducted an ablation study on Llama-3.1-8B-Instruct comparing three architecture sizes for the Size Policy (Hidden dimensions: [1024], [1024, 256], and [1024, 512, 256]), the results are reported in Table 10. The results show a significant performance jump when increasing from 1 to 2 layers. However, the marginal gain from 2 to 3 layers is minimal. Therefore, we selected the 2-layer architecture ([1024, 256]) for size policy to strike the optimal balance between training simplicity and inference efficiency.

## B  TRAINING DATA GENERALIZATION

To evaluate the generalization capabilities of our model across diverse domains, we employed the MMLU (Hendrycks et al., 2020) benchmark, a widely adopted dataset covering 57 tasks ranging from elementary mathematics and US history to computer science and law. We compared the speedup

Table 9: Latency and overhead analysis of dynamic policies on Vicuna-13B and Llama31-8B. Policy latency is measured in milliseconds (ms), total generation latency in seconds (s), and overhead in percentage (%).

| Time per Sample | MT-bench | Alpaca | Qa | Gsm8K | Mean |
|---|---|---|---|---|---|
| *Llama-3.1-8B-Instruct* | | | | | |
| Size Policy Latency (ms) | 21.4 | 6.22 | 6.36 | 5.83 | 9.96 |
| Depth Policy Latency (ms) | 40.8 | 12.4 | 10.2 | 11.9 | 18.8 |
| Total Latency per Sample (s) | 4.48 | 1.28 | 1.27 | 1.24 | 2.07 |
| Size Policy Overhead (%) | 0.478 | 0.485 | 0.500 | 0.469 | 0.483 |
| Depth Policy Overhead (%) | 0.910 | 0.969 | 0.803 | 0.955 | 0.909 |
| Total Overhead (%) | 1.39 | 1.45 | 1.30 | 1.42 | 1.39 |
| *Vicuna-13B-v1.3* | | | | | |
| Size Policy Latency (ms) | 13.7 | 4.92 | 4.90 | 5.56 | 7.26 |
| Depth Policy Latency (ms) | 27.9 | 9.61 | 8.39 | 12.0 | 14.5 |
| Total Latency per Sample (s) | 3.59 | 1.16 | 1.10 | 1.37 | 1.81 |
| Size Policy Overhead (%) | 0.381 | 0.423 | 0.444 | 0.406 | 0.414 |
| Depth Policy Overhead (%) | 0.777 | 0.826 | 0.761 | 0.874 | 0.810 |
| Total Overhead (%) | 1.16 | 1.25 | 1.20 | 1.28 | 1.22 |

Table 10: Speedup of the Size Policy across three architecture configurations on Llama-3.1-8B-Instruct.

| Configuration | MT-bench | Gsm8k | Alpaca | Qa | Mean |
|---|---|---|---|---|---|
| [1024] | 3.84 | 3.99 | 4.39 | 3.22 | 3.86 |
| [1024, 256] | 3.95 | 4.03 | 4.44 | 3.24 | 3.92 |
| [1024, 512, 256] | 3.96 | 4.07 | 4.43 | 3.25 | 3.93 |

ratios of our proposed method, LTD, against the Eagle3 baseline on the Llama-3.1-8B-Instruct model, as illustrated in Figure 7. The results demonstrate that LTD outperforms Eagle3 in 54 out of the 57 domains, underscoring the robust generalization of our approach across different fields. In particular, LTD achieves a speedup improvement of over $10\%$ in mathematics and logic-related tasks. It is worth noting that among the top five domains where LTD achieves the most significant gains over Eagle3, the baseline exhibits below-average speedup performance. This indicates that our method effectively compensates for the deficiencies of Eagle3 in specific domains. Furthermore, the efficacy of our approach is not limited to scenarios where Eagle3 underperforms; LTD maintains superior acceleration even in domains such as Human Aging and Astronomy, where Eagle3 already demonstrates performance significantly above its average.

## C DETAILS OF REINFORCEMENT LEARNING

### C.1 ANALYSIS OF TRAINING STEPS

To validate the appropriateness of our training budget, we analyze the relationship between the total training timesteps and the coverage of the dataset (HumanEval). Taking the Llama model as an example, the HumanEval dataset contains a total of approximately $N_{\text{total}} = 27829$ new tokens (ground truth).

#### C.1.1 STEPS FOR DEPTH POLICY

For the depth policy, the total training budget is set to $B_{\text{depth}} = 1,000,000$ steps. In our reinforcement learning (RL) formulation, one step corresponds to a single draft action or a verification action.

However, speculative decoding is stochastic; not all drafted tokens are accepted. To estimate the effective number of dataset traversals (epochs), we must account for the acceptance rate $\alpha$. Assuming a moderate acceptance rate (e.g., $\alpha \approx 0.7$), the number of draft steps required to generate the full

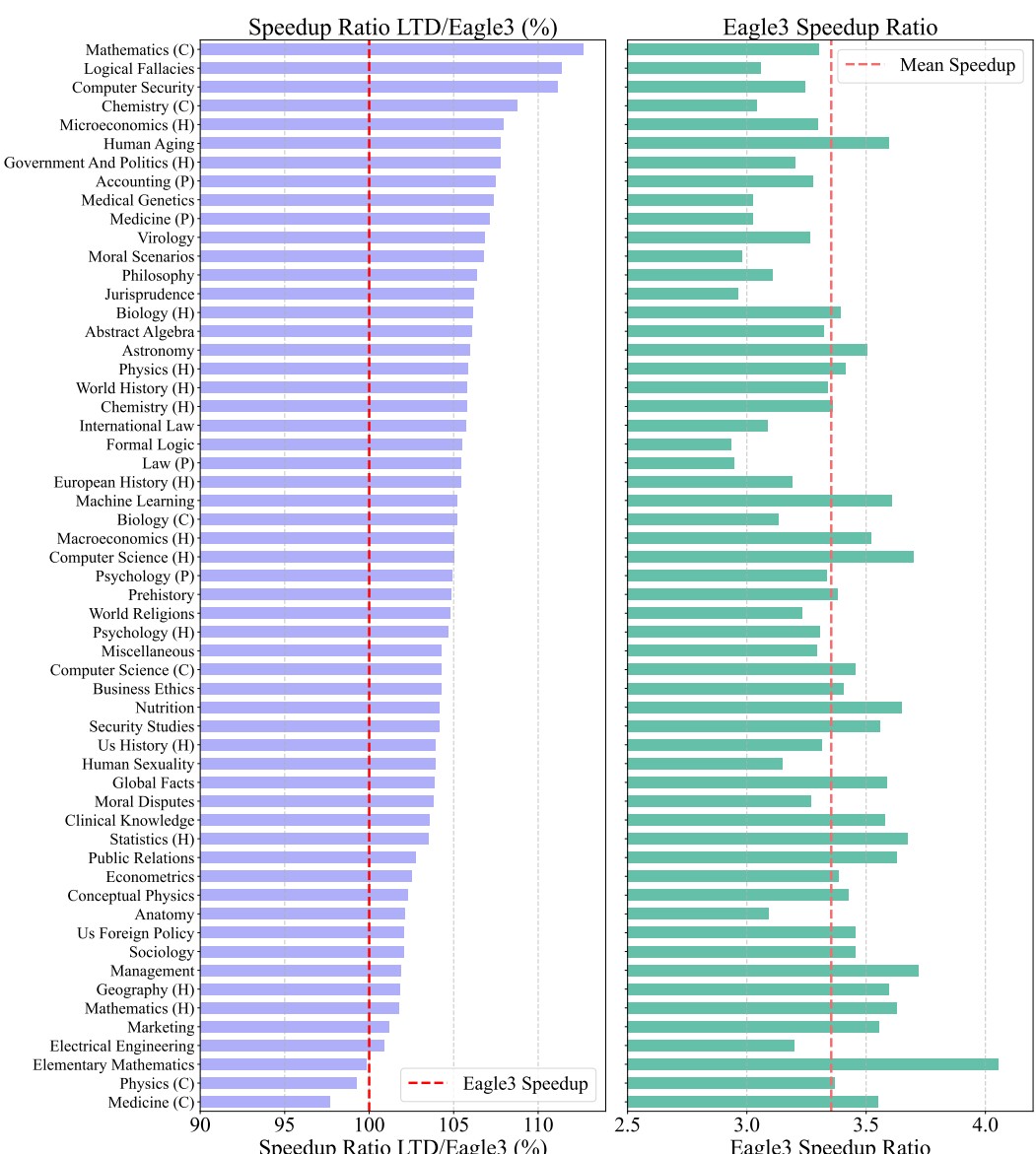

Figure 7: Speedup comparison between LTD and Eagle3 on the Llama-3.1-8B-Instruct model. We evaluated performance across 57 subtasks from MMLU(e.g., mathematics, history, law), using 80 sampled examples per subtask. Domain levels are abbreviated as H (High School), C (College), and P (Professional). Left: The relative speedup of LTD normalized against Eagle3; values greater than 100% indicate LTD outperforms Eagle3. Right: The absolute speedup ratio of the Eagle3 baseline across different subtasks.

dataset is greater than the number of accepted tokens:

$$\text{Step}_{\text{epoch}} \approx \frac{N_{\text{total}}}{\alpha} = \frac{27,829}{0.7} \approx 39,756 \text{ steps} \tag{6}$$

Consequently, the number of dataset traversals $K_{\text{depth}}$ is calculated as:

$$K_{\text{depth}} = \frac{B_{\text{depth}}}{\text{Step}_{\text{epoch}}} \approx \frac{1,000,000}{39,756} \approx 25.1 \text{ passes} \tag{7}$$

In an ideal scenario where $\alpha \to 1$ (or assuming 1 step $\approx$ 1 new token as a lower bound estimate for steps), the coverage would be approximately $1,000,000/27,829 \approx 35.9$ passes. Thus, our setting of 1M steps corresponds to roughly **25–40 traversals** of the dataset.

Given that the depth policy must learn to identify optimal termination points across varying contexts, and considering that PPO is an on-policy algorithm requiring repeated sampling of trajectories, this magnitude of exploration is standard and necessary for convergence, rather than excessive.

### C.1.2 STEPS FOR SIZE POLICY

For the size policy, the budget is set to $B_{\text{token}} = 100,000$ steps. This policy determines the total draft budget (action space size: 12 configurations, ranging from 20 to 240 tokens). A decision is made only once per speculative cycle.

Assuming an average acceptance length of 5 tokens per cycle, the policy is invoked once every 5 generated tokens. The steps required to traverse the dataset once are:

$$\text{Step}_{\text{epoch}} \approx \frac{N_{\text{total}}}{\tau} = \frac{27,829}{5} \approx 5,566 \text{ steps} \tag{8}$$

The total dataset traversals $K_{\text{token}}$ is:

$$K_{\text{token}} = \frac{B_{\text{token}}}{\text{Step}_{\text{epoch}}} \approx \frac{100,000}{5,566} \approx 18.0 \text{ passes} \tag{9}$$

This results in approximately **18–20 traversals**. Considering the policy must explore a discrete action space of 12 distinct configurations to optimize the trade-off between latency and generation length, this sampling volume is conservative and efficient for stabilizing the policy gradient.

### C.2 CONVERGENCE OF THE THROUGHPUT-BASED REWARD

To explicitly illustrate the dynamics of our throughput-based reward during training, we report the reward curves and validation speedup ratios for the Vicuna-13B size and depth policies, as shown in Figure 3. As observed, the rewards for both policies increase steadily as the number of training steps increases, and our allocated training budget is sufficient to ensure convergence. Regarding performance on the validation set, we observe an initial improvement followed by a declining trend. To prevent overfitting to the training set, we select the checkpoint that achieves the highest speedup on the validation set as our final model.

### C.3 TRAINING TIME COST

Due to the lightweight nature of the two policies, the RL training phase is highly efficient. For instance, training the Size Policy (100k steps) requires only 2.0, 2.5, and 3.5 hours for Llama3-8B, Vicuna-13B, and Qwen3-32B, respectively. Similarly, the Depth Policy (1M steps) takes 5, 5, and 7 hours on the same models. All experiments were conducted on a single NVIDIA A100 GPU. Even with three iterations of training, the total computational cost remains below 30 GPU hours. In practice, these policies are trained once, whereas decoding dominates the runtime in real-world deployments; thus, the marginal training overhead is effectively amortized by the improved inference efficiency. Given that LTD improves the speedup ratio by up to $36.4\%$ over Eagle-3, this additional training cost represents a highly favorable trade-off for latency-sensitive applications.

### C.4 PPO SELECTION

We deliberately select PPO since it can be integrated naturally into the online speculative decoding process, allowing us to record rewards during standard inference steps. Other RL methods, such as the currently popular GRPO, are not suitable for our task. We know GRPO requires repeated sampling from the same state to calculate baselines. In the context of speculative decoding, this would necessitate either repeatedly reconstructing the KV cache for different samples (computationally expensive) or generating multiple continuations sequentially (which introduces timing mismatches with the real inference environment). Therefore, we select PPO to train our policies.

## D  LTD EFFECTIVENESS ON GRIFFIN

To further validate the effectiveness of our approach across different speculative decoding methods, we selected Griffin (Hu et al., 2025), another method based on the dynamic tree framework. Griffin

Table 11: Speedup ratios and average acceptance lengths $\tau$ of different methods on Griffin using Meta-Llama-3-8B-Instruct during greedy decoding.

| Method | MT-bench | | Gsm8k | | Alpaca | | Qa | | Mean | |
| --- | --- | --- | --- | --- | --- | --- | --- | --- | --- | --- |
| | Speedup | $\tau$ | Speedup | $\tau$ | Speedup | $\tau$ | Speedup | $\tau$ | Speedup | $\tau$ |
| Griffin | 1.83 | 4.82 | 3.18 | 5.16 | 2.98 | 4.79 | 2.20 | 3.96 | 2.55 | 4.68 |
| Griffin+GS | 2.81 | 5.11 | **3.19** | 5.47 | 2.95 | 5.04 | 2.18 | 4.10 | 2.78 | 4.93 |
| Griffin+LTD | **2.88** | 4.98 | 3.13 | 5.45 | **2.98** | 5.01 | **2.23** | 4.10 | **2.81** | 4.89 |

features a distinct training loss and draft model architecture compared to Eagle3, but with a slightly inferior speedup. Using Meta-Llama-3-8B-Instruct (Dubey et al., 2024), we compared the speedup ratios of the default setting of GRIFFIN against our LTD method. The results in Table 11 demonstrate that our method achieves the state-of-the-art speedup on Griffin as well.

# E STATEMENTS

## USE OF LARGE LANGUAGE MODELS (LLMS)

The use of LLMs were strictly limited to language polishing, such as improving grammar, refining sentence structure, and enhancing the overall readability of the text. All suggestions and modifications proposed by the LLMs were critically reviewed and manually edited by the authors. This process ensured that the final text accurately and faithfully represents authors' original ideas, arguments, and findings. The core intellectual contributions of this work—including the research ideation, experimental design, data collection, analysis, and interpretation of results—are exclusively the work of the human authors. The LLM played no role in these critical research activities. The authors take full responsibility for all content presented in this paper, including the accuracy of the data and the validity of the conclusions.

