# OpenReview forum: "Learning To Draft: Adaptive Speculative Decoding with Reinforcement Learning"
_ICLR.cc/2026/Conference — ICLR 2026 Poster_

### Official Review · Reviewer_YsHJ · 2025-10-26

**Soundness:** 2
**Presentation:** 3
**Contribution:** 2
**Rating:** 4
**Confidence:** 3

**Summary:**

This work proposes Learning to Draft (LTD), a novel speculative decoding method that directly optimizes for throughput using reinforcement learning. LTD trains two distinct policies -- a depth policy for controlling the tree depth in drafting and a size policy for deciding the number of candidate tokens to verify -- that are co-optimized to jointly improve drafting and verification. Experimental results using the Eagle3 framework shows that LTD can yield substantial throughput improvements for greedy decoding across the Eagle3 baseline with various model families and sizes.

**Strengths:**

1. The paper's main premise, which is that optimizing for acceptance length is suboptimal, is well explain and is empirically supported.

2. The proposed RL method, which jointly trains two policies to directly optimize throughput, is intuitive and demonstrates promising empirical results for greedy decoding. The initial policy training and iterative optimization offer practical implementation benefits.

3. Experiments with multiple variants of the Eagle3 baseline with five different LLMs.

**Weaknesses:**

1. Missing discussion on training cost. While using sufficiently well trained policies have shown to often yield substantial throughput improvements, it is unclear if the cost of RL training is justified in all cases. Can one grid search be more efficient at the end than having to do RL training (which involves initial policy training + iterative optimization), especially for those cases where LTD doesn't lead to improvements (e.g., Dpsk 8B, Qwen3 14B, Qwe3 32B on several of the benchmarks)

2. Missing analysis on inference overhead. The lightweight MLP policies may add only "negligible overhead", more discussion on inference cost of running the policies in isolation would be useful. Also, do MLP policies with a greater capacity give better predictions? If so, what are the trade-offs between using larger MLP policies for better predictions vs. inference overhead. How do the answers change for different target models?

3. Training data generalization. It is somewhat surprising that the policies trained on HumanEval data give substantial improvements for non-coding benchmarks. There needs to be more discussion on the potential limits of generalization. Does training on more in-domain data lead to better results? How do policies trained on non-coding data transfer to other types of benchmarks?

**Questions:**

Q. How do the methods compare in case of sampling (as opposed greedy)?

Q. The amount of compute needed to train the policies seems to heavily depend on the target model. Have the authors done any analysis on potential policy generalization, i.e., how the two policies trained for a smaller target possibly generalize to a larger target?

---

> ### Author Response · Authors · 2025-11-22
> **Thanks for your valuable feedback!**
>
> We sincerely thank Reviewer YsHJ for the thoughtful feedback. We are encouraged that you recognize the value of our method, particularly our direct optimization of system throughput and our promising empirical results. Below, we address your concerns regarding training cost, inference overhead, and training data generalization.
>
> **Discussion on training cost**
>
> We agree that discussing the training cost is essential for a comprehensive evaluation.Therefore, we have included a detailed analysis of training costs in ​**Appendix C.3**​. All experiments were conducted on a single NVIDIA A100 GPU. Training the Size Policy (100k steps) requires only 2.0 to 3.5 hours across models ranging from Llama-3-8B to Qwen-32B, while the Depth Policy (1M steps) takes approximately 5 to 7 hours. Even with three iterations of co-adaptive training, the total computational cost remains below 30 GPU hours.
>
> In comparison, performing a standard Grid Search on a single Hummaneval dataset takes approximately 4 hours (testing 50 configurations: depths 4–12 and token limits 40–240, taking \~5 minutes per setting). Crucially, a static configuration derived from Grid Search lacks generalization across different data distributions. In contrast, our method is not merely a dynamic grid search; it dynamically selects the optimal speculation tree structure for *each* specific speculation step, resulting in superior adaptability.
>
> We further emphasize that our method achieves state-of-the-art speedup on ​DeepSeek-8B​, ​Qwen3-14B​, and Qwen3-32B across all test sets under greedy decoding. Regarding the acceptance length, we clarify that it is an auxiliary metric; the wall-clock speedup is the definitive measure of system efficiency.
>
> **Analysis on inference overhead**
>
> Inference latency of the MLP policies is indeed a critical factor. To quantify this, we have added ​**Table 9 in Appendix A.7**​, which details the time breakdown of our MLP models on Llama3.1-8B and Vicuna-13B. As illustrated below, the total time overhead of our policies is negligible: averaging only **1.4% on Llama3.1-8B** and ​**1.2% on Vicuna-13B**​. Since the Depth Policy is invoked more frequently, it incurs slightly higher overhead (\~0.8-0.9%) compared to the Size Policy (\~0.4-0.5%).
>
> | Time per Sample              | MT-bench | Alpaca | QA | GSM8K | Mean |
> | ------------------------------ | ---------- | -------- | ---- | ------- | ------ |
> | Llama-3.1-8B-Instruct        |
> | Size Policy Latency (ms)     | 21.4 | 6.22 | 6.36 | 5.83 | 9.96 |
> | Depth Policy Latency (ms)    | 40.8 | 12.4 | 10.2 | 11.9 | 18.8 |
> | Total Latency per Sample (s) | 4.48 | 1.28 | 1.27 | 1.24 | 2.07 |
> | Size Policy Overhead (%)     | 0.478 | 0.485 | 0.5 | 0.469 | 0.483 |
> | Depth Policy Overhead (%)    | 0.91 | 0.969 | 0.803 | 0.955 | 0.909 |
> | Total Overhead (%)           | 1.39 | 1.45 | 1.3 | 1.42 | 1.39 |
> | Vicuna-13B-v1.3              |
> | Size Policy Latency (ms)     | 13.7 | 4.92 | 4.9 | 5.56 | 7.26 |
> | Depth Policy Latency (ms)    | 27.9 | 9.61 | 8.39 | 12 | 14.5 |
> | Total Latency per Sample (s) | 3.59 | 1.16 | 1.1 | 1.37 | 1.81 |
> | Size Policy Overhead (%)     | 0.381 | 0.423 | 0.444 | 0.406 | 0.414 |
> | Depth Policy Overhead (%)    | 0.777 | 0.826 | 0.761 | 0.874 | 0.81 |
> | Total Overhead (%)           | 1.16 | 1.25 | 1.2 | 1.28 | 1.22 |

---

> > ### Author Response · Authors · 2025-11-22
> > **Thanks for your valuable feedback!**
> >
> > **The size selection for MLP policies**
> >
> > We thank the reviewer for this constructive suggestion regarding the trade-off between MLP policy capacity and inference overhead. There is an inherent trade-off in MLP design: we aim for the accuracy gains from a larger MLP to outweigh the increased inference cost. To validate our design choice, we conducted an ablation study on Llama3.1-8B comparing three architecture sizes for the Size Policy (Hidden dimensions: `[1024]`, `[1024, 256]`, and `[1024, 512, 256]`). The results (below) show a significant performance jump when increasing from 1 to 2 layers. However, the marginal gain from 2 to 3 layers is minimal. Therefore, we selected the 2-layer architecture (`[1024, 256]`) for size policy to strike the optimal balance between training simplicity and inference efficiency.
> >
> > | Configuration    | MT-bench | Gsm8k | Alpaca | Qa   | Mean |
> > | ------------------ | ---------- | ------- | -------- | ------ | ------ |
> > | [1024]           | 3.84     | 3.99  | 4.39   | 3.22 | 3.86 |
> > | [1024, 256]      | 3.95     | 4.03  | 4.44   | 3.24 | 3.92 |
> > | [1024, 512, 256] | 3.96     | 4.07  | 4.43   | 3.25 | 3.93 |
> >
> > **Training data generalization**
> >
> > We emphasize that our model does not learn domain-specific semantics, but rather captures the probability distribution patterns of draft tokens. As evidenced by ​**Table 6**​, we do not utilize hidden states as input. We selected HumanEval for training because code generation exhibits high variance in difficulty (ranging from rigid syntax to complex logic), providing a diverse spectrum of draft token probability distributions for the policy to learn from. To verify generalization, we evaluated our method on ​MMLU​, which consists of 57 subtasks across diverse domains (results detailed in **Appendix B** and ​**Figure 6**​).
> >
> > Our method outperforms Eagle3 on ​54 out of 57 subtasks​, achieving an average speedup improvement of ​5%​. Crucially, the speedup ranking is uncorrelated with the domain topic. For instance, the top-10 performing categories vary widely, including not only ​*Math*​, ​*Logic*​, but also ​*Human Aging*​, and ​*Government & Politics*, which is far from HumanEval. These results strongly support the cross-domain generalization of our approach.
> >
> > **Method Comparison under Sampling**
> >
> > To evaluate performance under sampling, we have compared our method at temperature T=1.0 (results listed in ​**Table 5**​). Notably, our method maintains the highest average speedup at a temperature of 1.0, despite being trained exclusively on greedy decoding. In contrast, other dynamic methods that adjust draft depth or verification size—while effective at greedy decoding—exhibit significant performance degradation at higher temperatures. This highlights the strong robustness of our method.
> >
> > **Policy Transfer to Larger Target Models**
> >
> > Since our input features are independent of the model's internal structure, transferring policies trained on small target models to larger  target models is technically feasible. However, we identify two critical issues with this approach. First, regarding verification latency mismatch, the optimal decision for the Size Policy is intrinsically coupled with the verification time of the specific target model. Consequently, a verification size that is efficient for a small model is unlikely to remain optimal for a larger model. Furthermore, we face the challenge of distribution shifts, where the distribution of token probabilities often differs across model scales. This discrepancy can potentially impair the policy's ability to accurately determine the optimal stopping criteria on a larger target model. Given that our training cost is very low (< 30 GPU hours even for Qwen3-32B), we recommend training directly on the target model to ensure optimal performance.
> >
> > ---
> >
> > We hope these clarifications regarding the training cost, inference overhead, and training data generalization address your concerns. We believe LTD offers a practical and robust solution for accelerating LLM inference. We would be grateful if you would consider raising your score based on these clarifications. Thank you again for your valuable review.

---

### Official Review · Reviewer_MWEV · 2025-10-29

**Soundness:** 3
**Presentation:** 3
**Contribution:** 3
**Rating:** 6
**Confidence:** 3

**Summary:**

This paper proposes Learning to Draft (LTD), a reinforcement learning-based speculative decoding method that directly optimizes throughput, defined as the number of accepted tokens per unit time in this paper, by jointly training two co-adaptive policies to control draft depth and verification size. Unlike prior static or heuristic approaches (e.g., Eagle3), LTD dynamically balances drafting and verification costs, achieving up to 36.4% speedup across five LLMs and four benchmarks while maintaining robust performance under different decoding settings.

**Strengths:**

- The paper introduces a principled reinforcement learning framework that directly optimizes throughput rather than proxy metrics like acceptance length, addressing a key limitation in prior speculative decoding methods.
- Extensive experiments across multiple LLMs and tasks demonstrate consistent and significant speedups (up to 36.4%) over strong baselines such as Eagle3. The analyses are also comprehensive and well-designed.
- The method generalizes well to high-temperature decoding scenarios where other speculative decoding methods typically degrade.

**Weaknesses:**

- The RL-based policy training (on HumanEval for 100K and 1M PPO steps) is expensive and tuned on a specific dataset. I am concerned about the training cost and the transferability to unseen domains or longer-context tasks. Could you provide the training details, including the actual GPU hours and why training for such extremely long steps?
- I am also concerned about the iterative optimization part. From Table 7 and 8, we can see that some models/datasets benefit from iterative optimization, while others demonstrate marginalized effect. The orders for optimization (size first or depth first) also lacks investigation. It would be great if the authors could provide some guidance on the design choices for iterative optimization, so that LTD will be more useful in practice.

**Questions:**

- In Figure 3, any intuitions as to Llama saturates while DeepSeek continues to gain advantage from iterative training?
- Some typos: (i) Make sure to add spacing before left brackets, e.g., lines 153 and 190. (ii) Figure 3 right, "token" should be "size".

---

> ### Author Response · Authors · 2025-11-22
> **Thanks for your valuable feedback!**
>
> We sincerely thank Reviewer MWEV for the thoughtful feedback. We are encouraged that you recognize the core value of our method, particularly our reinforcement learning framework that directly optimizes throughput and our method's robustness across varying temperatures. Below, we address your concerns regarding training costs, transferability, and design choices for iterative optimization.
>
> **Training Steps and GPU Hours**
>
> We are sorry for the confusion brought to you. While a training budget of "1 million steps" may appear substantial, it is important to contextualize this within our RL framework, a "step" corresponds to a single token action or verification decision, rather than a full dataset epoch. As detailed in our added analysis in **Appendix C.1**, given the HumanEval dataset size (approx. 27.8k tokens) and the typical acceptance rate, 1M steps for the Depth Policy corresponds to only 25--40 traversals of the dataset. Similarly, 100k steps for the Size Policy corresponds to roughly 18--20 traversals. Given that PPO is an on-policy algorithm requiring repeated sampling to stabilize gradients, this magnitude of exploration is standard and necessary for convergence, rather than excessive.
>
> To further justify this, we have visualized the training and validation rewards in **Figure 7**. Specifically, the Size Policy reaches its peak training reward around 100k steps. In contrast, while the Depth Policy appears to converge early on the training set (roughly 200k steps), its validation reward continues to improve, peaking  at approximately 700k steps. This indicates that our training steps are not too substantial for our dataset.
>
> Regarding the actual computational cost, the training phase is highly efficient due to the lightweight nature of the policies. On a single NVIDIA A100 GPU, training the Size Policy takes only 2.0 to 3.5 hours, and the Depth Policy takes 5 to 7 hours, depending on the base model. Even with three full iterations of co-adaptive training, the total cost remains below 30 GPU hours. This is a minor one-time cost compared to the long-term inference savings.
>
> **Transferability to unseen domains**
>
> We emphasize that our model does not learn domain-specific semantics, but rather captures the probability distribution patterns of draft tokens. Notably, we do not utilize hidden states as input, as evidenced in **Table 6**. To verify transferability to unseen domains, we evaluated our method on MMLU, which consists of 57 subtasks across diverse domains (added results detailed in **Appendix B and Figure 6**). Our method outperforms Eagle3 on 54 out of 57 subtasks, achieving an average speed improvement of 5%. Crucially, the speedup ranking is uncorrelated with the domain topic. For instance, the top-10 performing categories vary widely, including not only *Math* and *Logic* but also *Human Aging* and ​*Government & Politics*​—domains that are semantically distant from the code-centric HumanEval dataset. These results strongly support the cross-domain transferability of our approach.
>
> **Design Choices for Iterative Optimization**
>
> Regarding the order of optimization, we empirically investigated the training order and found that prioritizing the Size Policy yields better initial stability. We compared the average speedup on four benchmarks (MT-bench, GSM8K, Alpaca, QA) using Llama-3.1-8B when starting with either policy. As shown in the table below, training the Size Policy first resulted in a higher average speedup (3.98) compared to training the Depth Policy first (3.91). We attribute this to the fact that in the initial training, the size policy interact with a random depth baseline. Training the Size Policy first stabilizes the verification budget,  providing a more consistent environment for the Depth Policy to subsequently learn optimal termination points.
>
> | Configuration              | MT-bench | GSM8K | Alpaca | QA   | Mean |
> | ---------------------------- | ---------- | ------- | -------- | ------ | ------ |
> | Iter 0 + Train Depth Policy | 3.83     | 3.78  | 4.45   | 3.59 | 3.91 |
> | Iter 0 + Train Size Policy  | 3.99     | 3.89  | 4.49   | 3.54 | 3.98 |

---

> > ### Author Response · Authors · 2025-11-22
> > **Thanks for your valuable feedback!**
> >
> > **Intuition on Llama vs. DeepSeek Saturation**
> >
> > Regarding the difference in convergence behavior shown in Figure 3, we hypothesize that this stems from the probability distribution patterns of draft tokens. Llama3.1 likely exhibits a more distinct probability distribution where the optimal drafting strategy is easier to identify, allowing the policy to converge quickly and saturate after a single iteration. In contrast, DeepSeek may possess a more complex probability distribution, requiring the policies to undergo additional rounds of co-adaptation to find the optimal balance between draft depth and verification size. In practice, we recommend ​**2-3 iterations**​, as marginal gains typically diminish beyond this point. Given the low training cost (\~5 hours per iteration), this is computationally feasible for practical deployment.
> >
> > **Typos and Presentation**
> >
> > We thank the reviewer for pointing out the formatting issues. We have corrected the spacing before brackets (lines 153 and 190) and fixed the label in Figure 3 (changing "token" to "size") in the revised manuscript.
> >
> > ---
> >
> > We hope these clarifications regarding the training steps, transferability, and design choices for iterative optimization address your concerns. We believe LTD offers a practical and robust solution for accelerating LLM inference. We would be grateful if you would consider revisiting your assessment based on the new information.

---

### Official Review · Reviewer_c2Bw · 2025-11-01

**Soundness:** 3
**Presentation:** 4
**Contribution:** 3
**Rating:** 4
**Confidence:** 3

**Summary:**

This paper presents Learning to Draft (LTD), a reinforcement learning (RL) framework for optimizing speculative decoding efficiency in large language models.
Unlike prior methods, which use fixed or heuristic strategies to adjust the draft depth and verification size, LTD directly maximizes throughput (accepted tokens per total time) rather than surrogate metrics such as acceptance length.
The authors model the drafting–verification process as an RL environment with two co-adaptive policies: a Depth Policy (controls how deep the draft tree expands), and a Size Policy (controls how many candidate tokens are verified).
Both are trained via PPO using throughput as a direct reward.
LTD achieves 2.24×–4.32× speedup across five LLMs and four benchmarks, improving upon EAGLE-3 by up to 36.4% without quality degradation, and maintains robustness even under high-temperature sampling.

**Strengths:**

- The paper formalizes speculative decoding as a reinforcement learning (RL) environment that directly optimizes throughput rather than proxy metrics such as acceptance length. It introduces two interacting policies (Depth and Size), and the co-adaptive training framework is conceptually clear and empirically validated.
- The algorithmic structure is clearly explained, and the figures effectively illustrate the RL formulation and the draft–verify cycle.
- The proposed method demonstrates robustness across different sampling temperatures and models, showing consistent performance improvements.
- The approach can enhance real-world LLM inference speed without retraining or modifying the base model, which makes it practically valuable.

**Weaknesses:**

- The RL formulation lacks theoretical guarantees or convergence analysis under the throughput-based reward.
- The paper does not report quantitative metrics related to training efficiency (e.g., sample efficiency, or convergence speed).
- The paper does not discuss the rationale behind selecting PPO as the optimization algorithm for policy learning, nor whether alternative RL methods were considered.
- While the ablation studies demonstrate co-adaptation effects, the analysis remains qualitative. A more quantitative study (e.g., correlation between policy actions or mutual information) could strengthen the argument for policy synergy.
- The policies are trained on the HumanEval code dataset but evaluated on text-based benchmarks (e.g., GSM8K, MT-bench). The rationale or analysis explaining this generalization behavior is missing.
- The experimental section lacks comparisons with other adaptive or self-speculative methods such as DEL, or LayerSkip which would provide a stronger baseline context.

**Questions:**

- Can you report sample efficiency or convergence statistics (e.g., number of steps to reach stable reward or variance across runs)?
- What was the rationale for selecting PPO as the optimization algorithm? Were alternative RL methods considered?
- Have you analyzed the interaction strength between the Depth and Size policies (e.g., correlation of their actions during co-training)?
- What motivated the choice of HumanEval as the primary training environment?
- How much additional inference-time latency is introduced by running the two policy networks?

---

> ### Author Response · Authors · 2025-11-22
> **Thanks for your valuable feedback!**
>
> We sincerely thank Reviewer c2Bw for the thoughtful feedback. We are encouraged that you recognize the core value of our method, particularly our direct optimization of system throughput and the robustness across different sampling temperatures and models. We appreciate your careful inspection regarding training convergence, the rationale for our algorithmic choices, the interaction and latency of the policies and comparisons with self-speculative methods. We will address these concerns below.
>
> **Regarding ​Training Convergence and Sample Efficiency**
>
> To explicitly illustrate the dynamics of our throughput-based reward during training and address the concern regarding convergence, we have added the reward curves and validation speedup ratios for the Vicuna-13B size and depth policies in ​**Figure 7 of Appendix C**​. As observed, the rewards for both policies increase steadily as the number of training steps increases. Specifically, the Size Policy typically requires 20k–80k steps to reach reward stability, while the Depth Policy requires approximately 200k–400k steps. Our allocated training budget is sufficient to ensure convergence. Regarding performance on the validation set, we observe an initial improvement followed by a potential declining trend if training continues indefinitely. To prevent overfitting, we select the checkpoint that achieves the highest speedup on the validation set (e.g., for Vicuna's depth policy, this peak typically occurs around 700k steps).
>
> **Regarding ​PPO Selection**
>
> Regarding the optimization algorithm, we deliberately select PPO since it can be integrated naturally into the online speculative decoding process, allowing us to record rewards during standard inference steps. Other RL methods, such as the currently popular GRPO, are not suitable for our task. We know GRPO requires repeated sampling from the same state to calculate baselines. In the context of speculative decoding, this would necessitate either repeatedly reconstructing the KV cache for different samples (computationally expensive) or generating multiple continuations sequentially (which introduces timing mismatches with the real inference environment). We would follow your suggestion to include the selection rationale in the paper for clarity.
>
> **Analysis of Policy Interaction**
>
> To validate the effectiveness of our co-adaptive training and quantify the interaction between policies, we analyzed the test set distributions of the initial separately trained policies ('Iter0') and the final iteratively trained LTD policies, as visualized in ​**Figure 5 in Appendix A.6**​. We examined the relationship between the number of candidate tokens and the draft depth, conditioned on the acceptance length. The 'Iter0' baseline exhibits a pattern regardless of acceptance length, predominantly clustering in regions with shallow depth (\$<6\$) and moderate token count (\$<60\$). In contrast, LTD demonstrates superior adaptability. For large acceptance lengths (indicating easier generation), a significant proportion (43\%) of cases utilize both a large token budget and a deeper tree. Conversely, when the acceptance length is small (indicating higher difficulty), LTD frequently adopts a 'shallow but wide' strategy (depth \$<6\$, tokens \$>60\$). This strategy of proactively deploying more candidate tokens effectively mitigates the reduction in acceptance length caused by increased difficulty, confirming that our policies have learned to synergize effectively.

---

> ### Author Response · Authors · 2025-11-22
> **Thanks for your valuable feedback!**
>
> **Humaneval Selection**
>
> We emphasize that our policies do not aim to learn domain-specific semantics, but rather captures the probability distribution patterns of draft tokens. As evidenced by Table 6, we do not utilize hidden states as input, but instead rely on token probabilities. We selected HumanEval for training because code generation exhibits high variance in difficulty (ranging from rigid syntax to complex logic), providing a diverse spectrum of draft token probability distributions for the policy to learn from.
>
> To verify generalization, we evaluated our method on MMLU, which consists of 57 subtasks across diverse domains (results detailed in **Appendix B and Figure 6**). Our method outperforms Eagle3 on 54 out of 57 subtasks, achieving an average speed improvement of 5%， which further confirms that the policies generalize well to text domains, as they learn  probability patterns rather than domain-specific features.
>
> **Inference Latency of Policy Networks**
>
> We analyzed the latency overhead of our depth and size policies on both Llama-3.1-8B-Instruct and Vicuna-13B-v1.3. As reported in the newly added **Table 9** the computational overhead introduced by our policies is minimal: the total overhead is less than 1.5\% on Llama-3.1 and even lower (1.2\%) on the larger Vicuna model. Notably, the latency of the depth policy is approximately twice that of the size policy. Although we utilize a lightweight single-layer network, the depth policy incurs a higher total cost due to its more frequent invocation at every step of the draft construction, compared to the size policy which is invoked once per cycle.
>
> **Comparison with Self-Speculative Decoding**
>
> Our work is positioned within the lineage of tree-based speculative decoding methods, such as Medusa, SpecInfer, and the Eagle family. We focused our comparisons on Eagle3, as it is the current state-of-the-art in this specific category, and demonstrating improvements over it provides the strongest evidence of our contribution to tree structure optimization. While methods like self-speculative decoding (e.g., DEL, LayerSkip) are valuable, they primarily focus on reducing memory footprint by utilizing less model layers, which represents an orthogonal direction to the structural optimization of draft trees. Therefore, a direct comparison with Eagle3 is more relevant to isolating the efficacy of our proposed LTD framework.
>
> ---
>
> Thank you for your constructive suggestions. We hope our response can effectively addresses your questions regarding convergence, policy interaction and latency, and training data selection. We have reflected the response above into the new version of the paper. We would be grateful if you would consider raising your score based on these clarifications. Thank you again for your valuable review.

---

> > ### Author Response · Authors · 2025-11-26
> > **Generalization to other tree-based speculative decoding methods**
> >
> > To further validate the generalizability of our approach across different speculative decoding frameworks, we selected **GRIFFIN** [1], another method based on the dynamic tree structure. GRIFFIN employs a distinct training loss and draft model architecture compared to Eagle3, albeit with slightly lower baseline speedup. Using Meta-Llama-3-8B-Instruct as base model, we compared the speedup ratios of GRIFFIN—under both its default and Grid Search configurations—against our LTD method. Notably, we trained LTD for only a single iteration on HumanEval. The results below demonstrate that our method achieves state-of-the-art speedup on GRIFFIN as well, effectively verifying the generalization capability of our approach across different speculative decoding methods.
> >
> > | **Method** | **MT-bench** | **GSM8K** | **Alpaca** | **QA** | **Mean** |
> > | ------------------ | -------------------- | ----------------- | ------------------ | -------------- | ---------------- |
> > | GRIFFIN          | 1.83               | 3.18            | 2.97             | 2.20         | 2.55           |
> > | GRIFFIN+GS       | 2.81               | **3.19**       | 2.95             | 2.17         | 2.78           |
> > | **GRIFFIN+LTD** | **2.88**          | 3.13            | **2.98**        | **2.23**    | **2.81**      |
> >
> > [1]  Griffin: Effective token alignment for faster speculative decoding. arXiv preprint arXiv:2502.11018, 2025.

---

### Official Review · Reviewer_WBfk · 2025-11-01

**Soundness:** 3
**Presentation:** 4
**Contribution:** 3
**Rating:** 6
**Confidence:** 3

**Summary:**

The paper introduces Learning to Draft (LTD), a novel method for accelerating large language model (LLM) inference using adaptive speculative decoding. Unlike previous methods that rely on static configurations or optimize for proxy metrics like acceptance length, LTD directly optimizes for the throughput of each draft-and-verify cycle, accounting for the crucial time cost of both the drafting and verification phases. The authors solve this problem using a Reinforcement Learning (RL) environment and train two distinct, co-adaptive policies: a depth policy to dynamically control the draft tree's depth (draft cost) and a size policy to manage the verification size (verification cost). These policies are jointly optimized to maximize the throughput, defined as the number of accepted tokens divided by the total time ($L_A / T_{total}$). Through extensive evaluations, LTD demonstrated significant throughput improvements of up to 36.4% over the state-of-the-art Eagle3 baseline across various LLMs and tasks, proving a more robust and efficient dynamic strategy for speculative decoding.

**Strengths:**

1. The proposed method LTD directly optimizes the practical system throughput ($L_A / T_{total}$), i.e., the number of accepted tokens divided by the total time. This ensures the strategy is genuinely maximizing inference speed while prior works mostly focused on indirect metrics like acceptance length.

2. The method uses a Reinforcement Learning (RL) environment to train two dynamic, co-adaptive policies: one for the draft tree's depth and one for the verification size. This dynamic approach allows the model to continuously adjust both the drafting cost and the verification cost for optimal performance in real-time.

3. LTD demonstrates significant and robust performance gains, achieving speedup ratios up to 4.32x and improving throughput by up to 36.4% over state-of-the-art baselines like Eagle3. This indicates a highly effective and generalizable strategy for accelerating LLM inference across various models and tasks.

**Weaknesses:**

1. Training the adaptive policies requires setting up and running a complex RL environment. This adds significant computational overhead and complexity during the training phase, which is a major barrier to adoption compared to simpler, non-adaptive speculative decoding methods.

2. The core of LTD relies on accurately modeling the time cost of both the drafting and verification phases to compute the throughput objective ($T_{total}$). If the real-world environment introduces variances or non-linearities in time cost that the model does not capture, the policies trained in the simplified environment may become sub-optimal in production.

3. Like all speculative decoding methods, LTD's performance still fundamentally relies on the quality of the small draft model to accurately predict future tokens. While the adaptive policies optimize the use of the draft model, they cannot compensate for a draft model that frequently generates incorrect tokens, which would lead to low acceptance rates and minimal speedup.

4. The paper only compares against Eagle3 as the base framework, but does not evaluate on other tree-based methods like SpecInfer or Medusa. The contribution would be strong if more comparisons with recent adaptive methods are provided.

**Questions:**

1. How does the computational cost of training and deploying the two co-adaptive policies using the RL environment compare to the total inference time savings achieved? Is the initial overhead of the RL training process outweighed by the long-term throughput gains in practical, high-volume production settings?

2. If the small draft model's quality degrades (e.g., due to a mismatch between its training data and the target task) how effectively can the adaptive depth and size policies mitigate the resulting low token acceptance rate to maintain a high throughput? What is the minimum acceptable acceptance rate before LTD's advantages over static methods disappear?

---

> ### Author Response · Authors · 2025-11-22
> **Thanks for your valuable feedback!**
>
> We sincerely thank Reviewer WBfk for the thoughtful feedback. We are encouraged that you recognize the core value of our method, particularly our direct optimization of system throughput and the robustness of our co-adaptive policies in continuously adjusting drafting and verification costs. Below, we address your concerns regarding training complexity,  time cost modeling, robustness to draft model degradation and baseline comparisons.
>
> **Training Cost and Complexity of the RL Environment**
>
> Thanks for your concerns.  The trade-off between training cost and the resulting performance gain is indeed a critical factor. In this work, our RL training phase is highly efficient due to the lightweight nature of our policies. In our implementation, on a single NVIDIA A100 GPU, training the Size Policy（100K steps）takes only 2.0 to 3.5 hours ranging from Llama3.1-8B to Qwen3-32B, while the Depth Policy (1M steps) takes approximately 5 to 7 hours. Even with three iterations of co-adaptive training, the total computational cost remains below 30 GPU hours. We think that such a one-time cost is acceptable. In practical settings, the decoding process dominates the lifecycle of an LLM service; thus, the marginal training overhead is effectively amortized by the substantial improvement in inference efficiency. Given that LTD improves the speedup ratio by up to 36.4\% over Eagle3, this one-time investment represents a highly favorable trade-off for latency-sensitive applications. Furthermore, we have fully adapted the RL environment for speculative decoding tasks to ensure it can be easily migrated to different models, and we are committed to open-sourcing our code for the community.
>
> **Robustness to Time Cost Variances and Non-linearities**
>
> We appreciate your concerns regarding time cost variances in real-world environment. As you mentioned, there are indeed  frequent variances or non-linearities in time cost (e.g., due to cache hits/misses), which may cause individual actions to deviate from the optimal outcome. However, we argue that our RL-based approach is inherently robust to these factors. First, during inference, our policies do not attempt to explicitly predict time cost; instead, they map the current state directly to the optimal actions (verification size and draft depth). Second, the RL training process naturally accounts for some variations. The variance in time cost acts as stochasticity in the environment. Since RL optimizes for the expected cumulative reward (throughput) over many sampling steps, the resulting policies learn to select actions that yield high throughput on average, implicitly accounting for the non-linearities and fluctuations encountered during inference.

---

> ### Author Response · Authors · 2025-11-22
> **Thanks for your valuable feedback!**
>
> **Performance under varying Draft Model Quality**
>
> We appreciate your concerns regarding draft model quality, which may indeed vary across different domains. For the explicit empirical validation of our method's robustness under varying draft model qualities, we evaluated LTD on the MMLU benchmark. MMLU consists of 57 subtasks across diverse domains, allowing us to simulate dynamic draft model quality. (The detailed results are presented in **Appendix B and Figure 6.**)  LTD outperforms Eagle3 in 54 out of 57 tasks, demonstrating strong generalization capabilities. Crucially, we observed that among the top five domains where LTD achieves the most significant gains, the baseline often exhibits below-average speedup performance. This indicates that our adaptive policies effectively compensate for the deficiencies of the draft model in difficult domains by dynamically adjusting the tree structure. For example, when encountering challenging scenarios, our method can allocate more tokens for verification, thereby increasing the final acceptance length. Furthermore, even in domains where Eagle3 already performs significantly above average (e.g., Human Aging and Astronomy), LTD maintains superior acceleration. In the rare cases where LTD slightly underperforms, Eagle3 already achieves high speedups, suggesting that a static configuration happens to align with the optimal strategy in those specific instances. In the vast majority of scenarios, however, the dynamic adaptability of LTD provides a crucial advantage.
>
> **Comparison with Other Tree-based Methods**
>
> We focused our primary comparison on Eagle3 because it represents the current state-of-the-art in tree-based speculative decoding. Methods like SpecInfer and Medusa rely on static tree structures (pre-determining which nodes to verify), which are incompatible with dynamic tree framework. Since LTD is designed as an algorithmic advancement upon the dynamic tree paradigm, demonstrating significant gains over Eagle3 establishes the superiority of our method. Regarding adaptive methods, we have conducted comparisons against recent approaches such as C2T, SpecDec++, and GammaTune, demonstrating the distinct advantages of our method. We trust that these comparisons effectively substantiate the effectiveness of our approach.
>
> ---
>
> We hope our response has effectively addressed your concerns regarding ​training costs, robustness to timing variances and generalization capabilities.  We believe LTD offers a practical and robust solution for accelerating LLM inference. We would be grateful if you would consider revisiting your assessment based on this new information.

---

> > ### Author Response · Authors · 2025-11-26
> > **Further Empirical Verification on Tree-based Methods**
> >
> > To further validate the generalizability of our approach across different speculative decoding frameworks, we selected **GRIFFIN** [1], another method based on the dynamic tree structure. GRIFFIN employs a distinct training loss and draft model architecture compared to Eagle3, albeit with slightly lower baseline speedup. Using Meta-Llama-3-8B-Instruct as base model, we compared the speedup ratios of GRIFFIN—under both its default and Grid Search configurations—against our LTD method. Notably, we trained LTD for only a single iteration on HumanEval. The results below demonstrate that our method achieves state-of-the-art speedup on GRIFFIN as well, effectively verifying the generalization capability of our approach across different speculative decoding methods.
> >
> > | **Method** | **MT-bench** | **GSM8K** | **Alpaca** | **QA** | **Mean** |
> > | ------------------ | -------------------- | ----------------- | ------------------ | -------------- | ---------------- |
> > | GRIFFIN          | 1.83               | 3.18            | 2.97             | 2.20         | 2.55           |
> > | GRIFFIN+GS       | 2.81               | **3.19**       | 2.95             | 2.17         | 2.78           |
> > | **GRIFFIN+LTD** | **2.88**          | 3.13            | **2.98**        | **2.23**    | **2.81**      |
> >
> > [1]  Griffin: Effective token alignment for faster speculative decoding. arXiv preprint arXiv:2502.11018, 2025.

---

### Author Response · Authors · 2025-11-30
**General Response (1)**

Dear PC, AC and Reviewers,

We sincerely thank all of you for your efforts and time on the reviewing.

We appreciate the constructive feedback and are particularly encouraged that the reviewers highlighted several key strengths of our work: our principled Reinforcement Learning framework, which directly optimizes system throughput rather than relying on suboptimal proxy metrics like acceptance length; our co-adaptive policy framework, where two policies learn jointly to dynamically coordinate for each phase; and our significant empirical performance gains (up to 36.4% improvement over Eagle3) with exceptional robustness across diverse LLMs, tasks, and high-temperature settings.

All four reviewers showed approval of the main ideas in our paper and they were all fine with the paper being accepted. We especially appreciate their constructive feedback on some specific details. In response, we have provided detailed explanations and conducted additional experiments to make the paper clearer and more solid. We also summarized our responses below:

**Regarding ​**​**Time cost and Training Details for RL:  ​**All reviewers suggested that we could report the training cost and provide more details regarding the RL training process. Accordingly, we clarify that our RL training phase is highly efficient due to the lightweight nature of our policies. In our implementation, on a single NVIDIA A100 GPU, training the Size Policy（100K steps）takes only 2.0 to 3.5 hours ranging from Llama3.1-8B to Qwen3-32B, while the Depth Policy (1M steps) takes approximately 5 to 7 hours (detailed in​**​ Appendix C.3**​). Even with three iterations of co-adaptive training, the total computational cost remains ​**below 30 GPU hours**​. We think that such a one-time cost is acceptable.   In practical settings, the decoding process dominates the lifecycle of an LLM service; thus, the marginal training overhead is effectively amortized by the substantial improvement in inference efficiency. Given that LTD improves the speedup ratio by up to 36.4% over Eagle3, this one-time investment represents a highly favorable trade-off for latency-sensitive applications.

To explicitly illustrate the dynamics of our throughput-based reward during training and address the concerns regarding convergence, we have added the reward curves and validation speedup ratios for the Vicuna-13B size and depth policies in ​**Figure 7 of Appendix C**​. Our allocated training budget is sufficient to ensure convergence. To prevent overfitting, we select the checkpoint that achieves the highest speedup on the validation set.   We provide detailed training configurations in the ​**Appendix C**​, and we will release our code as open source to facilitate reproduction by the community.

**Regarding ​**​**Selection of HumanEval and Training Data Generazliation:** Reviewers c2Bw, MWEV and YsHJ expressed concerns about the selection of HumanEval as the training dataset and suggested conducting supplementary experiments to verify the model's generalization ability in other domains. As suggested, we clarified the reason why we selected HumanEval for training: our policies aim to leverage the probability distribution patterns of draft tokens for prediction, and the code generation task in this benchmark exhibits high variance in difficulty (ranging from rigid syntax to complex logic), providing a diverse spectrum of draft token probability distributions for the policy to learn from. Critically, our approach is not tied to the code domain: as evidenced by Table 6, our policy relies solely on probability and token length as inputs, without incorporating any domain-specific semantic features like hidden states. This design ensures the model learns universal distribution patterns rather than overfitting to code semantics.

To verify generalization, we evaluated our method on MMLU, which consists of 57 subtasks across diverse domains (results detailed in **Appendix B** and ​**Figure 6**​). Our method outperforms Eagle3 on 54 out of 57 subtasks, achieving an average speedup improvement of 5%. Crucially, the speedup ranking is uncorrelated with the domain topic. For instance, the top-10 performing categories vary widely, including not only ​*Math*​, ​*Logic*​, but also ​*Human Aging*​, and ​*Government & Politics*​, which is far from HumanEval. These results strongly support the cross-domain generalization of our approach.

---

> ### Author Response · Authors · 2025-11-30
> **General Response (2)**
>
> ​**Regarding the Latency brought by two Policies**​**​:​**​**​ ​**Reviewers c2Bw and YsHJ requested an elaboration on the costs associated with the two policies in our method.  According to the suggestions, we show the time breakdown of our MLP policies on Llama3.1-8B and Vicuna-13B (results detailed in ​**​ Appendix A.7 and Table 9**​). As illustrated in the table, the total time overhead of our policies is negligible: averaging only **1.4% on Llama3.1-8B** and ​**1.2% on Vicuna-13B**​. Since the Depth Policy is invoked more frequently, it incurs slightly higher overhead (\~0.8-0.9%) compared to the Size Policy (\~0.4-0.5%).  We also conducted an analysis (detailed in**​ Appendix A.8 and Table10** ) demonstrating that our architecture size strikes a good balance between predictive capability and the corresponding latency.
>
> The suggestions have been invaluable in helping us refine our paper and highlight our fundamental contributions approved by the reviewers. We have carefully incorporated all the above clarifications into the new version of our paper. We hope that the clarifications above could address the reviewers' primary concerns and underscore the paper's contribution to accelerate LLM inference.

---

### Meta-Review · Area_Chair_ueYs · 2026-01-05

**Summary:**

This paper focuses on improving the efficacy of speculative decoding (SD). The authors  argue that drafting and verification are interdependent and should be co-optimized. Toward this goal, the authors propose Learning to Draft (LTD), an RL framework which seeks to directly optimize the throughput of each draft-and-verify cycle. LTD leverages two co-adaptive policies to  dynamically manage the allocation between draft generation and target verification: a depth policy that determines the draft tree’s depth, and a size policy for selecting the optimal number of candidate tokens to be verified by the target model.  These policies are optimized concurrently using throughput as the reward signal. The authors evaluate LTD for diverse LLMs and tasks, and observe speedup ratios that compare favorably with state of the art methods like Eagle 3.

**Reviewer Concerns:**

The reviewers expressed concern about the training cost and overhead due to the RL framework, and asked for additional training details, which the authors provided in their rebuttal, also arguing that the relatively small  training  overhead justifies efficiency gains during inference.  Three of the reviewers commented on  the choice of the training data and generalization properties of the learned policies. While the authors explanation of why they selected humaneval for training purposes was not completely satisfying, they provided additional experimental results on MMLU showing that Eagle3+LTD outperforms Eagle3 on 54 out of 57 subtasks, thus confirming its generalization ability. There was a related question about the robustness of LTD in  a realistic  environments, where the time cost of drafting and verification might be very different from those costs  in simplified scenarios, and thus policies learned on those scenarios might be suboptimal. The authors’ response stated that RL-based approach provides inherent robustness against such variations since (1) the policies do not explicitly predict time cost but map states to actions; (2) RL training process naturally accounts for some variations. This answer, especially the first part, does not seem very satisfying, because a systematically different time-cost behavior might lead to suboptimal state to action mapping.  Finally, one of the major concerns was the limited experimental scope and omission of certain baselines.. In the rebuttal, the actors provide additional results with GRIFFIN, an SD method that leverages  dynamic tree structure. Despite  the author’s assertion that LTD “..achieves state-of-the-art speedup on GRIFFIN..”, the results indicate a more mixed picture, where Griffin+GS outperforms LTD on one out of 4 datasets, and trails only slightly for the other three.

**Reviewer Scores:**

WBfk unchanged;
c2Bw 4->6;
MWEV unchanged;
YsHJ 4-> 6;

---

### Decision · Program_Chairs · 2026-01-26

Accept (Poster)